# Hepatitis B virus RNAs co-opt ELAVL1 for stabilization and CRM1-dependent nuclear export

Yingcheng Zheng[1,2☉], Mengfei Wang[1☉], Jiatong Yin[1], Yurong Duan[1], Chuanjian Wu[1], Zaichao Xu[1], Yanan Bu[1], Jingjing Wang[1], Quan Chen[1], Guoguo Zhu[3], Kaitao Zhao[1], Lu Zhang[1], Rong Hua[1], Yanping Xu[1], Xiyu Hu[1], Xiaoming Cheng[1,4,5,6], Yuchen Xia [1,6,7]*

**1** State Key Laboratory of Virology and Hubei Province Key Laboratory of Allergy and Immunology, Institute of Medical Virology, TaiKang Center for Life and Medical Sciences, TaiKang Medical School, Wuhan University, Wuhan, China, **2** School of Life Sciences, Hubei University, Wuhan, China, **3** Department of Emergency, General Hospital of Central Theater Command of People's Liberation Army of China, Wuhan, China, **4** Department of Pathology, Zhongnan Hospital of Wuhan University, Wuhan, China, **5** Hubei Clinical Center and Key Laboratory of Intestinal and Colorectal Diseases, Wuhan, China, **6** Hubei Jiangxia Laboratory, Wuhan, China, **7** Pingyuan Laboratory, Henan, China

☉ These authors contributed equally to this work.
* yuchenxia@whu.edu.cn

**Data Availability Statement:** All relevant data are within the manuscript and its Supporting information files.

**Funding:** This work was supported by the Open Grant from the Pingyuan Laboratory (2023PY-OP-

## Abstract

Hepatitis B virus (HBV) chronically infects 296 million people worldwide, posing a major global health threat. Export of HBV RNAs from the nucleus to the cytoplasm is indispensable for viral protein translation and genome replication, however the mechanisms regulating this critical process remain largely elusive. Here, we identify a key host factor embryonic lethal, abnormal vision, Drosophila-like 1 (ELAVL1) that binds HBV RNAs and controls their nuclear export. Using an unbiased quantitative proteomics screen, we demonstrate direct binding of ELAVL1 to the HBV pregenomic RNA (pgRNA). ELAVL1 knockdown inhibits HBV RNAs posttranscriptional regulation and suppresses viral replication. Further mechanistic studies reveal ELAVL1 recruits the nuclear export receptor CRM1 through ANP32A and ANP32B to transport HBV RNAs to the cytoplasm via specific AU-rich elements, which can be targeted by a compound CMLD-2. Moreover, ELAVL1 protects HBV RNAs from DIS3+RRP6+ RNA exosome mediated nuclear RNA degradation. Notably, we find HBV core protein is dispensable for HBV RNA-CRM1 interaction and nuclear export. Our results unveil ELAVL1 as a crucial host factor that regulates HBV RNAs stability and trafficking. By orchestrating viral RNA nuclear export, ELAVL1 is indispensable for the HBV life cycle. Our study highlights a virus-host interaction that may be exploited as a new therapeutic target against chronic hepatitis B.

## Author summary

Export of HBV RNAs from the nucleus to the cytoplasm is indispensable for viral protein translation and genome replication, however the mechanisms regulating this critical

0101), the National Natural Science Foundation of China (project no. 81971936), Hubei Province's Outstanding Medical Academic Leader Program, the Fundamental Research Funds for the Central Universities (project no. 2042022kf1215, 2042023kf0129 and 2042021gf0013), and Basic and Clinical Medical Research Joint Fund of Zhongnan Hospital, Wuhan University. The funders had no role in study design, data collection and analysis, decision to publish, or preparation of the manuscript.

**Competing interests:** The authors have declared that no competing interests exist.

process remain largely elusive. We demonstrate that ELAVL1 recognizes HBV RNAs and recruits ANP32A and ANP32B to transport viral RNA from nucleus to cytoplasm via CRM1 pathway. CMLD-2, a drug with ELAVL1 binding potential, restricts HBV replication. Functional cure of chronic HBV infection would require combination of current therapies with additional novel agents. Export of HBV RNAs from the nucleus to the cytoplasm is indispensable for viral protein translation and genome replication. Our study provides a proof of concept illustrating that targeting ELAVL1 holds promise as a potential approach to suppress HBV.

## Introduction

Hepatitis B virus (HBV) infection is a major public health threat, with an estimated 296 million people chronically infected [1]. Current approved therapies, interferon and nucleos(t)ide analogues, can prevent development of HBV related cirrhosis and hepatocellular carcinoma, but rarely eradicate virus infection [2–4]. HBV is a partially double-stranded DNA virus belonging to the *Hepadnaviridae* family. The viral particle envelopes with three viral surface antigens (HBsAg) that are named large (L), middle (M), and small (S). Viral nucleocapsid harbors HBV genome, a 3.2 kb relaxed circular DNA (rcDNA). HBV infection initiate with attaching to heparan sulfate proteoglycans [5,6] and interacting with viral receptor sodium taurocholate co-transporting polypeptide (NTCP) [7]. After entering the hepatocytes, HBV delivers rcDNA to the nucleus to form a minichromosome, called covalently closed circular DNA (cccDNA) [8,9]. cccDNA serves as a template for transcription of viral RNAs, including pregenomic RNA (pgRNA), pre-core RNA (pcRNA), preS1 RNA, preS2/S RNA and HBV X protein (HBx) mRNA [8,10–12]. Among them, pgRNA is the template for HBV core protein (HBc) and polymerase translation as well as reverse transcription template to form HBV DNA. The viral genome-containing nucleocapsids are either enveloped and released as progeny virions or transported back to the nucleus to amplify the cccDNA pool [2,13].

The nuclear export of HBV RNAs is an important process that allows virus to synthesize its proteins through translation machinery and replicate its genome through revers-transcription in the cytoplasm. The molecular mechanisms that underlie the HBV RNAs nuclear export remain largely obscure. It has been shown that transport of HBV RNAs from the nucleus to the cytoplasm is regulated by a posttranscriptional *cis* regulatory element (PRE) overlapping EN1 and a portion of the X gene [14]. By using plasmid containing partial HBV sequence, Chi *et al.* reported that PRE is necessary for TRanscription and EXport complex (TREX) recruitment on viral mRNAs and viral mRNA nuclear export [15]. In addition, NXF1-p15 machinery is reported to be involved in HBV pgRNA export [16]. However, there are evidences showing that PRE contribute to pgRNA stability, but has little effect on its nuclear export [17]. Beside naked viral RNAs, a recent paper demonstrated that CRM1 machinery can mediate nuclear export of capsid containing viral RNAs [18]. Nevertheless, as cytoplasmic HBV RNAs are essential templates for viral protein translation, nuclear export of encapsidated RNAs is unlikely the predominant pathway for viral RNAs export.

To dissect the molecular pathway governing nuclear export of HBV RNAs, we utilized an unbiased quantitative proteomics approach to screen for host proteins that interact with HBV RNA. This identified Embryonic lethal, abnormal vision, Drosophila-like 1 (ELAVL1, or HuR) as an HBV RNA binding partner. Further investigation revealed that ELAVL1 knockdown restricted HBV replication in both cell culture and mouse models. Mechanistic studies

disclosed that by binding AU-rich elements (AREs), ELAVL1 regulated both HBV RNA stability and nuclear export via the CRM1 pathway.

## Results

### ELAVL1 is an HBV proviral host factor

To elucidate the molecular mechanism underlying HBV RNAs nuclear export, we performed RNA pulldown coupled with quantitative proteomics to identify proteins interacting with pgRNA (Fig 1A). Pathway analysis of the 60 enriched factors (S1A Fig) revealed the ELAVL1--mediated RNA stabilization pathway (Fig 1B). Network analysis highlighted a central role of ELAVL1 and CRM1 (Fig 1C), which was further validated by Cluster analysis (S1B Fig). To investigate the role of ELAVL1 in HBV replication, we knocked down ELAVL1 in HBV-infected HepG2-NTCP cells and observed decreased HBV DNA, HBeAg and HBsAg in supernatant (Fig 1D). Intracellularly, ELAVL1 knockdown suppressed HBc-associated HBV DNA levels (Fig 1E) and HBc expression (Fig 1F) without affecting HBV cccDNA (Fig 1G). Further validation was conducted using primary human hepatocytes. Efficient knockdown of ELAVL1 (Fig 1H) resulted in inhibition of secreted antigens (Fig 1I), HBV DNA in supernatant (Fig 1I), and intracellular HBV RNA (Fig 1J). Similar results were obtained in HepAD38 cells, an HBV expressing cell line, and HBV infected HepG2-NTCP cells with stable ELAVL1 knockdown by shRNA (S2 Fig). Collectively, these data identify ELAVL1 as a proviral host factor that regulates HBV life cycle at the stage post cccDNA formation.

### ELAVL1 mediates HBV RNAs posttranscriptional regulation

To further elucidate the mechanism, we focused on the effects on HBV RNAs, as ELAVL1 is a potential viral RNA binding protein. ELAVL1 knockdown decreased HBV RNAs levels in both the cytoplasm and nucleus (Fig 2A). RNAscope confirmed that ELAVL1 knockdown reduced cytoplasmic and nuclear HBV RNAs levels without affecting the proportion of virus-positive cells (Fig 2B). Reconstitution of ELAVL1 expression in stable knockdown cells rescued levels of different HBV replication markers (S3A–S3C Fig). To further confirm the effect, we treated HBV infected HepG2-NTCP cells with CMLD-2, a known ELAVL1 inhibitor [19]. Similar to the effects of ELAVL1 knockdown, CMLD-2 treatment suppressed secreted HBV DNA, HBV antigens (Fig 2C) and intracellular HBc (Fig 2D) in a dose dependent manner. Cytoplasmic and nuclear HBV RNAs levels were also decreased by CMLD-2 (Fig 2E) without cytotoxicity (Fig 2F). Accordingly, a slight but significant inhibition of ELAVL1 HBV RNA interaction mediated by CMLD-2 was observed (Fig 2G). Similar HBV inhibition effects were observed in HepAD38 cells treated with CMLD-2 (S3D–S3G Fig). Next, we investigated whether ELAVL1 affects HBV RNAs at the transcriptional or posttranscriptional level. ELAVL1 knockdown did not affects the activity of HBV promotors and enhancers (Fig 2H), but accelerated pgRNA degradation in the presence of transcription inhibitor Actinomycin D (Fig 2I). Together, these results indicate that ELAVL1 is involved in the posttranscriptional regulation of HBV RNAs.

### Elavl1 regulates HBV RNAs *in vivo*

To further validate the role of ELAVL1 *in vivo*, we transduced mice with adeno-associated virus (AAV) expressing HBV as well as AAV encoding Elavl1 shRNA (Fig 3A). As expected, AAV-mediated knockdown of Elavl1 significantly reduced secreted viral antigens in the serum (Fig 3B) and HBc-associated DNA in the liver of the mice (Fig 3C) [20]. Western blotting verified efficient reduction of hepatic Elavl1 accompanied by inhibition of HBc expression (Fig

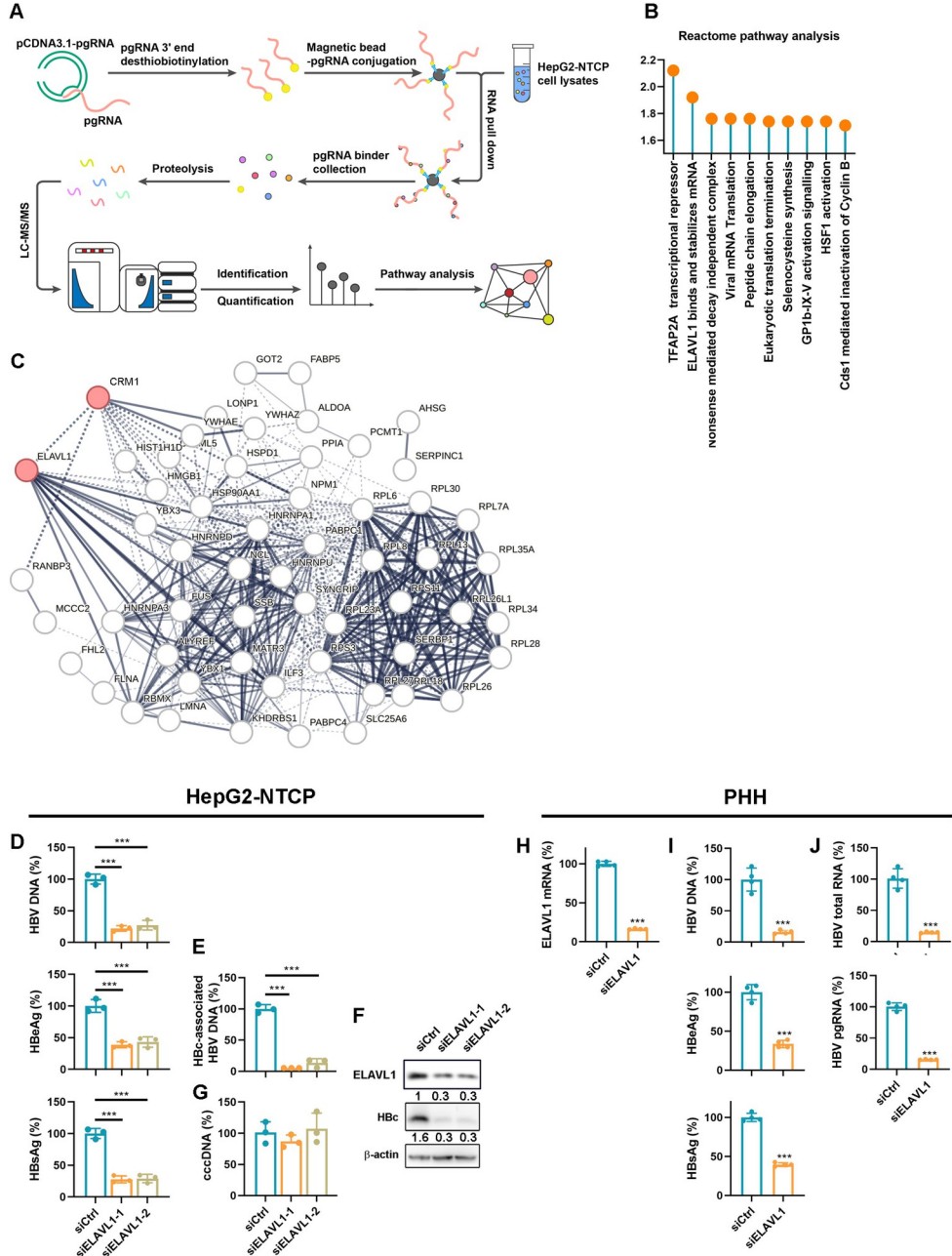

**Fig 1. ELAVL1 supports HBV replication.** (A) Work flow of the RNA pulldown-LC-MS/MS. (B) Reactome pathway analysis for the pgRNA binders. (C) Network analysis for the pgRNA binding proteins. (D-G) HepG2-NTCP cells were transfected with ELAVL1 targeted siRNA. The cells were infected with HBV at an MOI = 200 and were maintained with DMEM medium supplemented with 2.5% DMSO for 5 days. (D) Levels of HBV DNA in supernatants were determined by qPCR (% of siCtrl). Secreted HBeAg and HBsAg levels were determined by ELISA (% of siCtrl). (E and G) Levels of intracellular HBc-associated DNA and HBV cccDNA were determined by qPCR (% of siCtrl). (F) Levels of ELAVL1 and HBc were determined by WB. (H-J) The primary human hepatocytes were transfected with ELAVL1 targeted siRNA 2 days before HBV infection (MOI = 200). Samples were collected 5days post HBV infection. (H) Knockdown efficiency was evaluated by qPCR (% of siCtrl). (I) The levels of HBV DNA in culture supernatant were determined by qPCR (% of siCtrl). The levels of HBeAg and HBsAg in culture supernatant were determined by ELISA (% of siCtrl). (J) The levels of intracellular HBV RNA were assessed by qPCR. Graphs were shown as mean ± SD. *, p < 0.05; ***, p < 0.001.

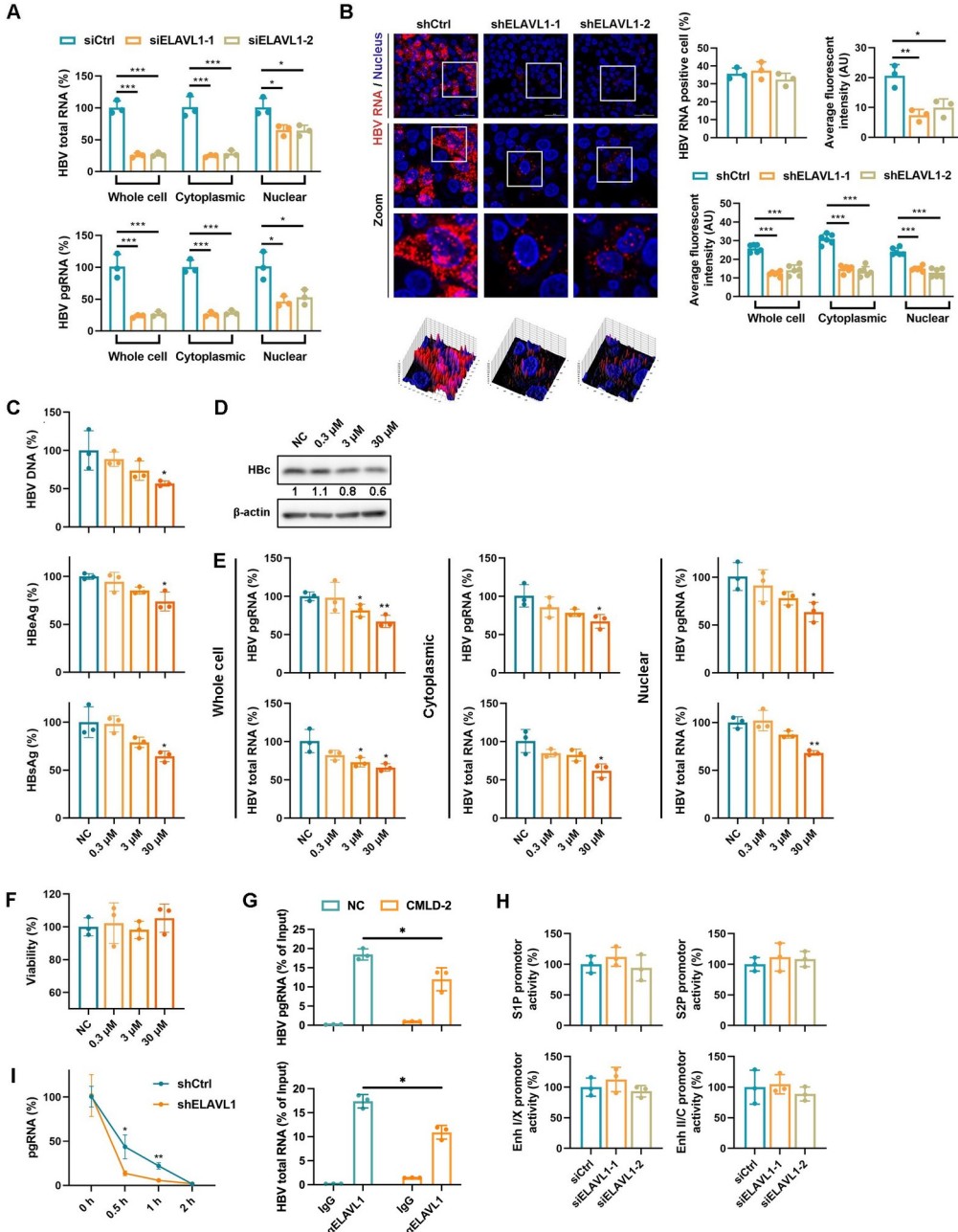

**Fig 2. ELAVL1 mediates HBV RNAs posttranscriptional regulation.** (A) HepG2-NTCP cells were transfected with ELAVL1 targeted siRNA. The cells were infected with HBV at an MOI = 200 and were maintained with DMEM medium supplemented with 2.5% DMSO for 7 days. Subcellular levels of HBV mRNA in cytoplasm and nucleus were determined by qPCR. Graphs were shown as mean ± SD. *, p < 0.05; ***, p < 0.001. (B) The ELAVL1 knockdown HepG2-NTCP-K7 cells or control cells were infected with HBV at an MOI = 200. The cells were harvested at 7 dpi. RNAscope assay was applied to observe the subcellular distribution of HBV RNAs. Graphs were shown as mean ± SEM. *, p < 0.05; **, p < 0.01; ***, p < 0.001. (C-F) HepG2-NTCP cells were infected with HBV at an MOI = 200 and were maintained with DMEM medium supplemented with 2.5% DMSO for 3 days. The cells were treated with CMLD-2 for 2 days as indicated. Graphs were shown as mean ± SD. *, p < 0.05; **, p < 0.01. (C) Levels of HBV DNA in supernatants were determined by qPCR (% of NC). Secreted HBeAg and HBsAg levels were determined by ELISA (% of NC). (D) Levels of HBc were determined by WB. (E) Subcellular levels of HBV mRNA in cytoplasm and nucleus were determined by qPCR. (F) Cell viability was evaluated by CCK-8 assay. (G) HepAD38 cells were treated with 30 μM CMLD-2 for 2 days. RIP assay was performed to analysis ELAVL1-HBV RNA binding in the cells. Levels of HBV RNA in ELAVL1 antibody immunoprecipitation were detected by qPCR. (H) Huh7 cells were transfected with ELAVL1 siRNAs 24 hours prior to transfection of the HBV promoter luciferase reporter plasmids

pGL-3-S1-promoter, pGL-3-S2-promoter, pGL-3-enhancer I/X-promoter, and pGL-3-enhancer II/precore/core-promoter respectively. Firefly luciferase activity (renilla luciferase activity as reference) was measured 48 hours after transfection (% of siCtrl). (I) Plasmid pCDNA3.1-pgRNA was transfected into Huh7 cells with ELAVL1 knockdown (shELAVL1). 48 hours post transfection, the cells were treated with 10 nM Actinomycin D for times as indicated. Levels of pgRNA were determined by qPCR (normalized to Cp value of pCDNA3.1-pgRNA, % of 0 hour).

3D). Immunohistochemistry revealed that Elavl1 knockdown markedly decreased intrahepatic HBs levels (Fig 3E). Mirroring cell culture observations, both cytoplasmic and nuclear HBV RNAs levels were reduced by Elavl1 knockdown in the liver (Fig 3F). RNAscope confirmed this finding (Fig 3G). Collectively, these data indicate that Elavl1 regulates HBV RNAs in AAV-HBV transduced mice.

## AU-rich elements in HBV RNAs are essential for ELAVL1 binding

As an RNA binding protein, ELAVL1 recognizes AREs in target RNAs [21–24]. We analyzed sequences from different HBV genotypes and found conserved canonical AREs (S4 Fig). Thus, we speculated that AREs were required for ELAVL1 binding to HBV RNA. We synthesized full-length HBV pgRNA with or without synonymous ARE mutations as baits (Fig 4A). RNA pulldown assay indicated that only wild type pgRNA interacted with ELAVL1 (Fig 4B). LC-MS/MS identification of the pulldown factors also showed that only wild type but not AREs mutant pgRNA interacted with ELAVL1 and CRM1 (Fig 4C). Using lysates of Myc-HBc transfected Huh7 cells expressing pgRNA as positive control, RNA immunoprecipitation (RIP) verified ELAVL1 binding to pgRNA (Fig 4D). To further validate this AREs dependent interaction, we synthesized 277 nt pgRNA probes, wild type or mutant, for RNA electrophoretic mobility shift assay (REMSA) (Fig 4E). Results showed ELAVL1 only bound wild type pgRNA probe, but not mutant (Fig 4F and 4G). Cold competition and super-shift REMSA confirmed this conclusion (Fig 4H). We next transfected Huh7 cells with plasmids expressing wild type or ARE-mutant pgRNA. ARE mutation in pgRNA reduced HBc protein expression and decreased cytoplasmic and nuclear pgRNA levels (Fig 4I and 4J) and accelerated pgRNA degradation (Fig 4K), suggesting AREs are critical for pgRNA regulation. To determine which HBV RNAs bind ELAVL1, we performed RIP using lysates of Huh7 cells transfected with plasmids expressing individual HBV RNA transcripts. The result confirmed ELAVL1 binds all HBV RNAs except HBx RNA (Fig 4L), which could be explained by the lack of AREs in HBx mRNA (Fig 4M). Together, these data indicate AREs in HBV RNAs are required for ELAVL1 recognition and regulation.

## CRM1 is a nucleocytoplasmic transport factor for HBV RNAs

CRM1, another host factor identified by our LC-MS/MS assay, is a known nuclear export receptor for ELAVL1 and its bound RNAs [24]. To investigate its role in HBV life cycle, we first confirmed the interaction of pgRNA and CRM1 by RNA pulldown (Fig 5A) and RIP assay (Fig 5B). Interestingly, ELAVL1 knockdown disrupted the interaction between CRM1 and HBV RNAs (Fig 5C) indicating that CRM1 indirectly interact with HBV RNAs via ELAVL1. Next, we transfected HepG2-NTCP cells with CRM1 siRNA, then infected them with HBV. Knockdown of CRM1 significantly decreased supernatant HBV DNA, secreted antigens, and intracellular HBc-associated DNA (Fig 5D and 5E) but had no effect on cccDNA levels (Fig 5F). Knockdown efficiency was validated by immunoblotting (Fig 5G). Interestingly, cell fractionation indicated that CRM1 deficiency selectively reduced cytoplasmic but not nuclear HBV RNAs (Fig 5H), which was further confirmed by RNAscope (Fig 5I). Similar results were obtained in stable CRM1 knockdown HepAD38 cells and HBV-infected

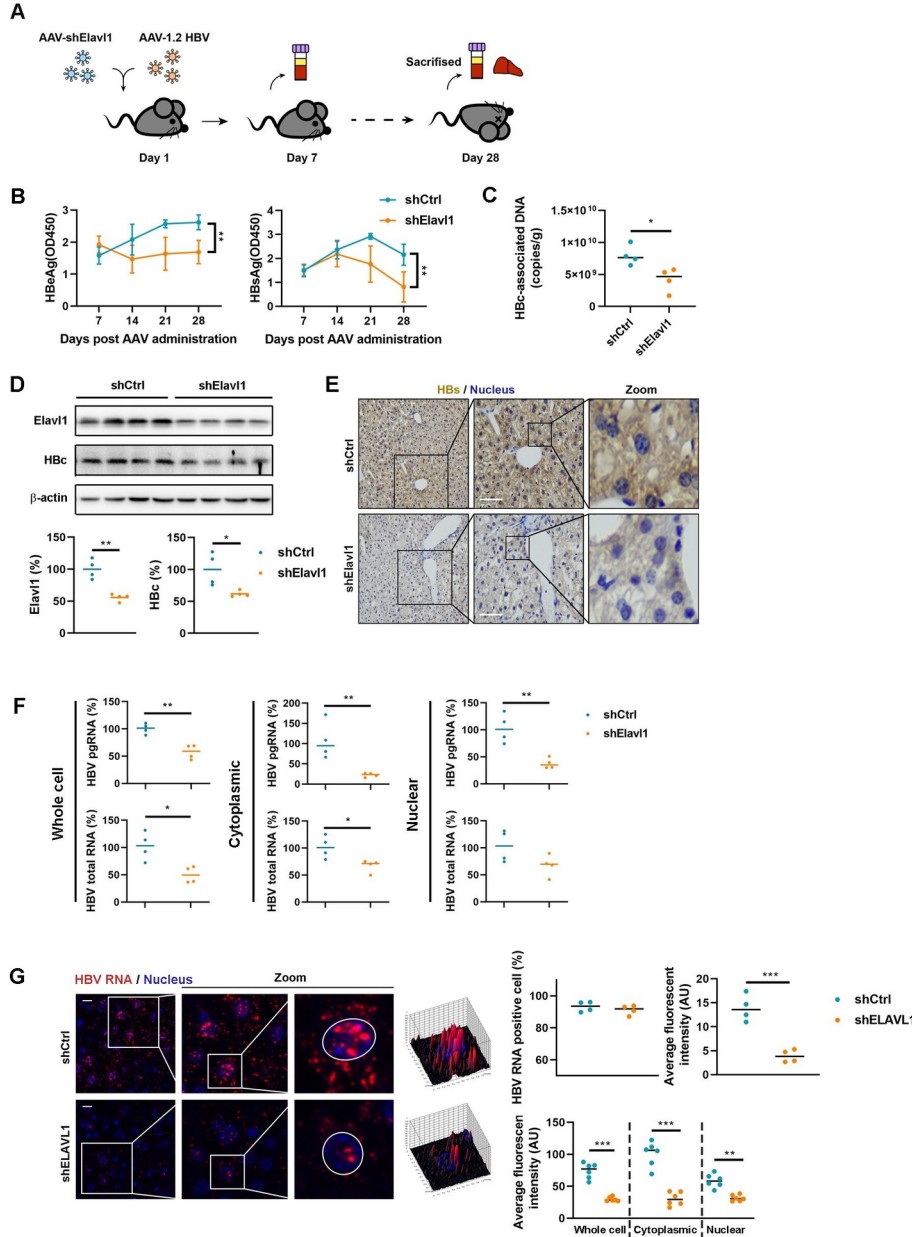

**Fig 3. Elavl1 regulates HBV RNA *in vivo*.** (A) Schematic description of the experiment workflow. The C57B/6 mice were transduced with AAV-1.2 HBV along with AAV expressing Elavl1-targeted shRNA. Blood was collected every 7 days. The mice were sacrificed 28 days post transduction. (B) Levels of HBeAg (1:50 dilution) and HBsAg (1:2000 dilution) in serum were determined by ELISA. (C) Levels of intrahepatic HBc-associated DNA were evaluated by qPCR. (D) Levels of intrahepatic Elavl1 and HBc were determined by WB. Intensity of the bands was determined by ImageJ software (% of shCtrl). (E) Intrahepatic HBs were detected by IHC. Bar = 50 μm. (F) Subcellular levels of intrahepatic HBV RNA in cytoplasm and nucleus were determined by qPCR (% of shCtrl). (G) RNAscope assay was applied to determine the distribution of intrahepatic HBV RNAs. Graphs were shown as mean ± SEM. *, p < 0.05; **, p < 0.01; ***, p < 0.001.

HepG2-NTCP cells (S5 Fig). These data suggest CRM1 regulates HBV replication by mediating RNA nucleocytoplasmic transport.

Beside CRM1, NXF1 is another major nuclear export receptor [25,26]. It has been reported that pgRNA and HBc protein export depend on NXF1-p15 pathway in an HBV plasmid

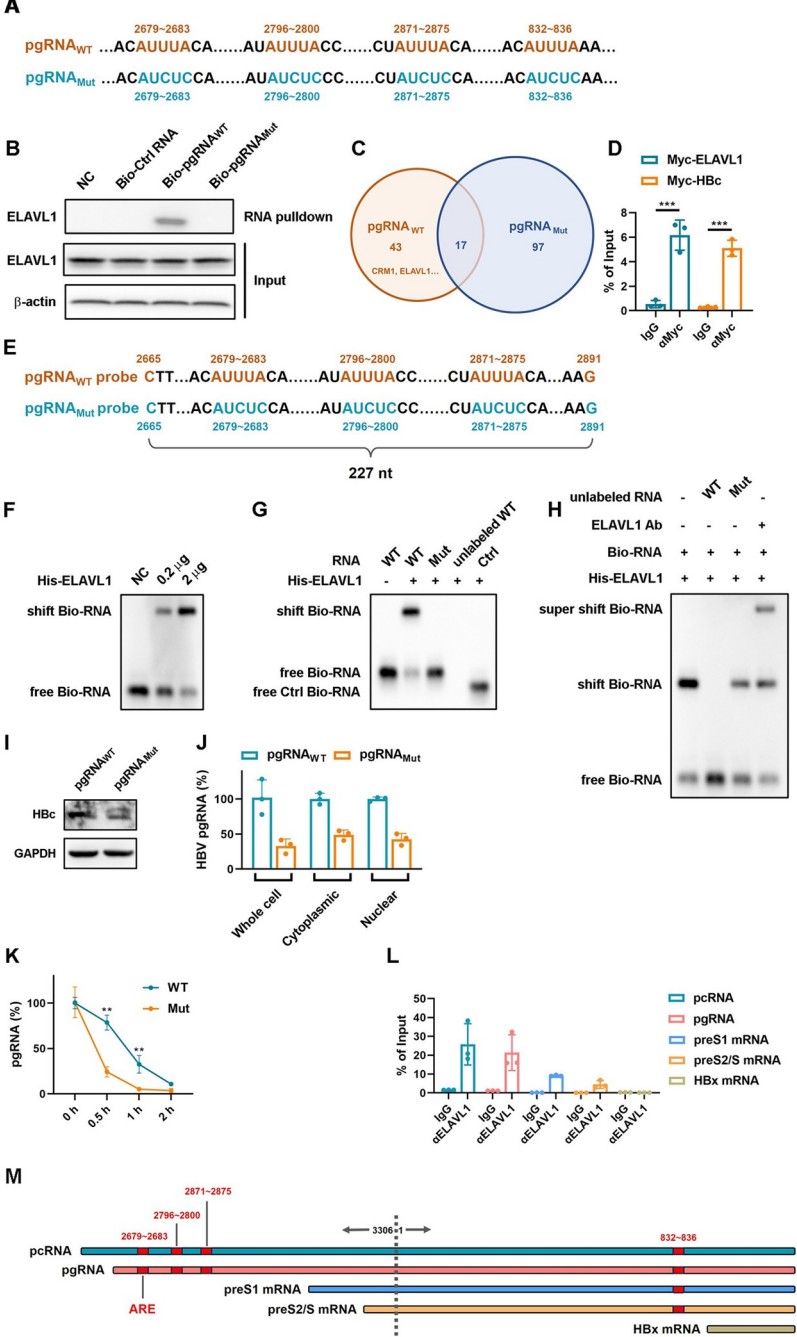

**Fig 4. AU-rich elements in HBV RNAs are essential for ELAVL1 binding.** (A) Sequence of wild type pgRNA and mutant pgRNA used for RNA pulldown. (B) HepG2-NTCP cell lysates were incubated with biotinylated pgRNA or its mutant coated magnetic beads to pull down binding proteins. Levels of ELAVL1 in beads elutes were detected by WB. (C) Venn diagram of pgRNA and its mutant binders identified by RNA pulldown-LC-MS/MS. (D) Myc-ELAVL1 and Myc-HBc were co-transfected with pCDNA3.1-T7-pgRNA into Huh7 cells respectively. RIP assay was conducted to investigate interaction of ELAVL1-pgRNA and HBc-pgRNA. (E) Schematic diagram of pgRNA probe (227nt) and its mutant used for REMSA. (F) *In vitro* expressed and purified ELAVL1 were incubated with biotin labeled pgRNA probes. The samples were subjected to REMSA assay to detect the pgRNA probe and ELAVL1 binding. (G) The biotin labeled pgRNA$_{AUUUA\ to\ AUCUC}$ probe or wild type pgRNA probe was incubated with in vitro purified ELAVL1. REMSA assay was performed to detected pgRNA and ELAVL1 binding. Unlabeled wild type pgRNA probe and biotin labeled IRE1 probe as control. (H) Before biotin labeled pgRNA probe incubation, unlabeled pgRNA probe or the mutant was incubated with ELAVL1 (for cold compete). Before biotin labeled pgRNA probe incubation, the purified ELAVL1 was incubated with ELAVL1 targeted antibody (for super shift assay). The samples were subjected to REMSA

assay for biotin labeled pgRNA probe and ELAVL1 binding detection. (I and J) Huh7 cells were transfected with pgRNA or its mutant (mutation site as indicated in A) expressing plasmid. (I) Levels of HBc expression were determined by WB. (J) Subcellular levels of HBV mRNA in cytoplasm and nucleus were determined by qPCR (% of WT). (K) The cells were treated with 10 nM Actinomycin D for times as indicated. Levels of pgRNA were determined by qPCR (normalized to Cp value of pCDNA3.1-pgRNA$_{WT}$ or pCDNA3.1-pgRNA$_{Mut}$, % of 0 hour). (L) The plasmids expressing pcRNA, pgRNA, PreS1 mRNA, PreS2/S mRNA, and HBx mRNA were transfected into Huh7 cells respectively. RIP assay was conducted to investigate their binding ability to ELAVL1. (M) Sequence analysis for AREs in HBV RNAs.

transfection model [16]. To clarify the roles of NXF1 and CRM1 in HBV replication, we applied NXF1-targeted shRNAs (S6A Fig). Similar to the previous report, knockdown of NXF1 in HBV-infected HepG2-NTCP cells reduced the secreted HBV DNA and antigens levels (S6B Fig) as well as intracellular HBV RNAs (S6C Fig). However, RIP assay revealed that NXF1 interact with its known target HSP70 [27] mRNA, but not HBV RNAs (S6D Fig). Notably, NXF1 knockdown suppressed ELAVL1 and CRM1 expression (S6E Fig) but ELAVL1 knockdown had no effect on NXF1 levels (S6F Fig). These data suggest NXF1 indirectly regulates HBV replication by orchestrating ELAVL1 and CRM1 expression, rather than directly binding and exporting HBV RNAs.

Leptomycin B (LMB), a potent antifungal antibiotic, has been discovered to inactivate CRM1 by covalent modification of the Cys528 residue [28]. To further confirm that CRM1 regulates HBV RNAs nucleocytoplasmic transport through recognition of nuclear export signals, we tested the effects of an LMB-resistant C528S mutant CRM1 on HBV RNAs nuclear export under LMB treatment. RIP assay indicated that C528S mutation did not affect pgRNA-CRM1 interaction (Fig 5J). However, complementation with CRM1-C528S, but not wild-type CRM1, made HBc expression resistant to LMB treatment (Fig 5K), as well as secreted HBV markers (Fig 5L) and cytoplasmic HBV RNAs levels (Fig 5M). These results indicate that LMB-mediated inhibition of CRM1 prevents nuclear export of HBV mRNAs and viral protein expression. The resistance of the C528S mutant confirms that CRM1 regulates HBV RNA nucleocytoplasmic transport through recognition of nuclear export signals.

## HBc is dispensable for HBV RNAs nuclear export

Recently, Yang *et al.* reported that CRM1 mediated nuclear export of capsid containing HBV RNAs [18]. To investigate whether HBc is necessary for HBV RNAs export, we transfected Huh7 cells with AAV-HBV1.2ΔHBc plasmid harboring a 1.2-fold HBV genome deficient in HBc expression (Fig 6A). RIP assay indicated that CRM1 bound to pgRNA independently of HBc (Fig 6B). Subcellular fractionation and RNAscope analysis revealed no difference in cytoplasmic and nuclear HBV RNAs levels under HBc-deficient conditions (Fig 6C and 6D). These data indicate HBc is not required for HBV RNAs nuclear export. We further tested whether ELAVL1-mediated HBV RNAs nuclear export relies on HBc. We transfected AAV-HBV1.2ΔHBc plasmid into Huh7 cells with ELAVL1 knockdown. HBc expression and ELAVL1 knockdown were confirmed by immunoblotting (Fig 6E). HBc deficiency had no effect on secreted antigen levels (Fig 6F) or cytoplasmic and nuclear HBV RNAs levels (Fig 6G). Overall, these data indicate HBc is dispensable for CRM1/ELAVL1 HBV RNA interaction or viral RNA nucleocytoplasmic transport.

## ANP32A and ANP32B bridge HBV RNA-ELAVL1-CRM1 and mediate HBV RNAs nuclear export

Besides ELAVL1, other two adaptors NXF3 [29] and EIF4E [30] could also mediate CRM1-dependent RNA nuclear export. However, neither NXF3 nor EIF4E knockdown affected HBV

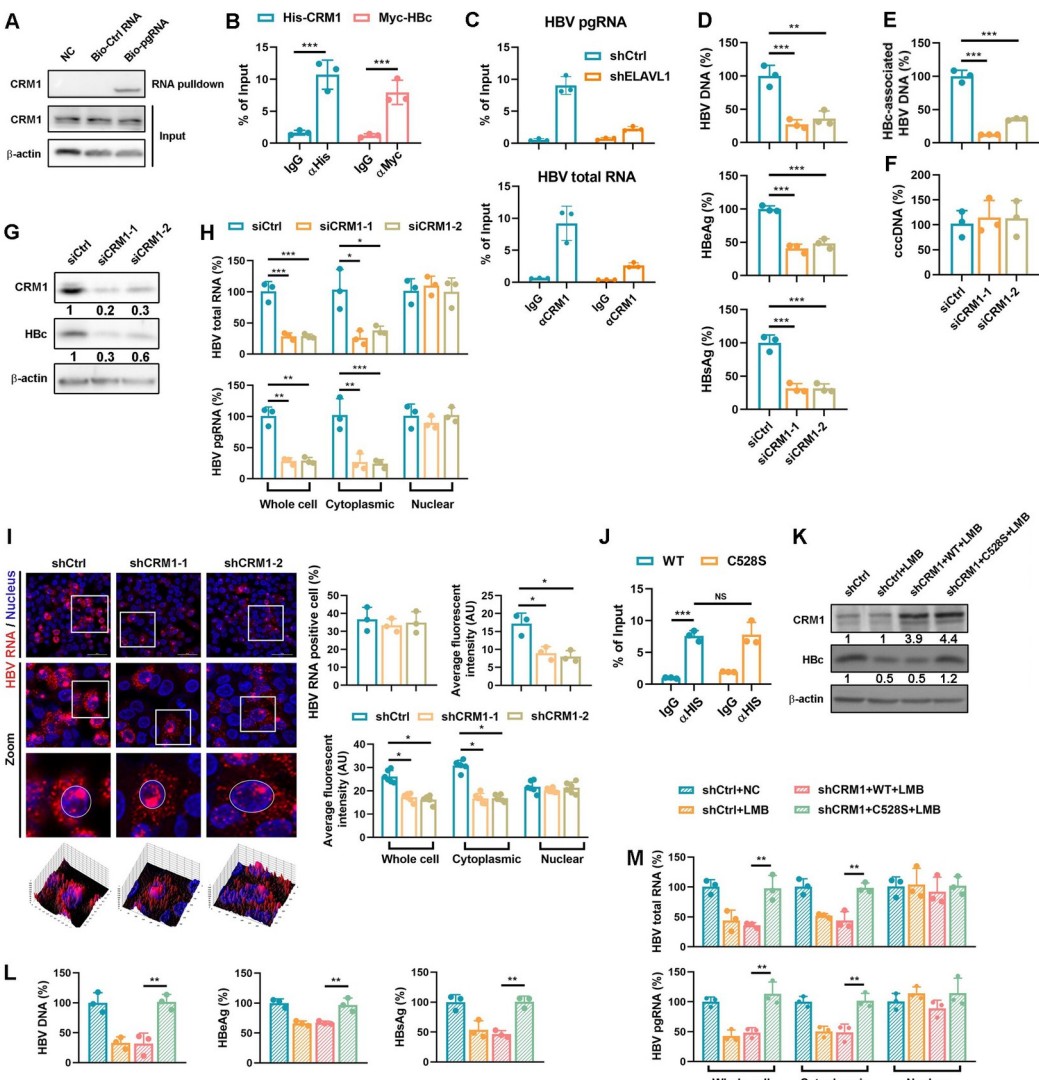

**Fig 5. CRM1 is a nucleocytoplasmic transport factor for HBV RNAs.** (A) HepG2-NTCP cell lysates were incubated with wide-type or mutant biotinylated pgRNA or its mutant coated magnetic beads to pull down binding proteins. Levels of CRM1 in beads elutes were detected by WB. (B) His-CRM1 and pcDNA3.1-pgRNA were transfected into Huh7 cells. CRM1-pgRNA interaction was determined by RIP-qPCR. Myc-HBc and pcDNA3.1-pgRNA transfected Huh7 cells was set as a positive control. (C) ELAVL1 knockdown HepAD38 cells were maintained in DMEM medium containing 2% DMSO for 2 days. The binding of HBV RNA-CRM1 were detected by RIP assay. (D-H) HepG2-NTCP cells were transfected with CRM1 targeted siRNA. The cells were infected with HBV at an MOI = 200 and were maintained with DMEM medium supplemented with 2.5% DMSO for 5 days. (D) Levels of HBV-DNA in supernatants were determined by qPCR (% of siCtrl). Levels of secreted HBeAg and HBsAg levels were determined by ELISA (% of siCtrl). (E and F) Levels of intracellular HBc-associated DNA and HBV cccDNA were determined by qPCR (% of siCtrl). (G) Levels of CRM1 and HBc were determined by WB. (H) Subcellular levels of HBV mRNA in cytoplasm and nucleus were quantified by qPCR (% of siCtrl). Graphs were shown as mean ± SD. *, $p < 0.5$; **, $p < 0.01$; ***, $p < 0.001$. (I) The CRM1 knockdown HepG2-NTCP-K7 cells or control cells were infected with HBV at an MOI = 200. The cells were harvested at 7 dpi. RNAscope assay was applied to evaluate the subcellular distribution of HBV RNAs. Graphs were shown as mean ± SEM. *, $p < 0.05$; **, $p < 0.01$; ***, $p < 0.001$. Data were shown as mean ± SEM. *, $p < 0.05$. (J) His-CRM1 or its C528S mutant was transfected with pcDNA3.1-pgRNA into Huh7 cells. CRM1-pgRNA interaction was detected by RIP. (K-M) CRM1 knockdown HepG2-NTCP cells were transfected with wild type CRM1 or its C528S mutant. The cells were infected with HBV at an MOI = 200 and were maintained with DMEM medium supplemented with 2.5% DMSO for 7 days. The cells were treated with 250 ng/ml Leptomycin B from 3 days post infection. (K) Levels of CRM1 and HBc were determined by WB. (L) Levels of HBV-DNA in supernatants were determined by qPCR (% of shCtrl). Levels of secreted HBeAg and HBsAg were determined by ELISA (% of shCtrl). (M) Subcellular levels of HBV mRNA in cytoplasm and nucleus were determined by qPCR (% of shCtrl). Graphs were shown as mean ± SD. **, $p < 0.01$; ***, $p < 0.001$.

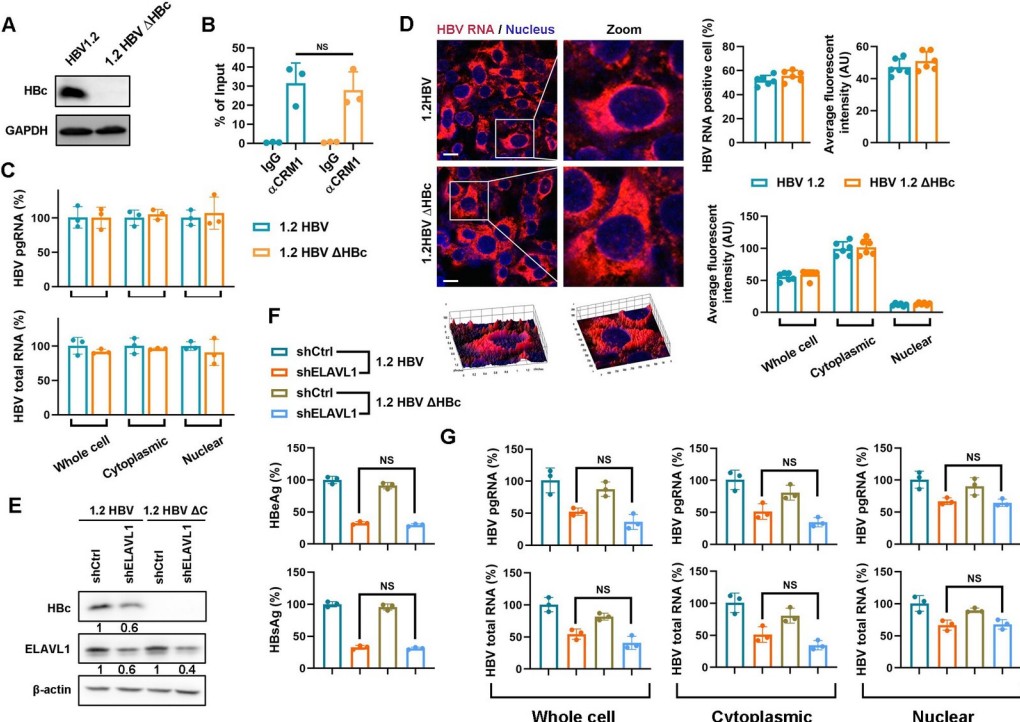

**Fig 6. HBc is dispensable for HBV RNAs nuclear export.** (A-D) 1.2 HBV or 1.2 HBV ΔHBc construct was transfected into Huh7 cells. (A) HBc deletion was confirmed by WB. (B) The binding of pgRNA-CRM1 were detected by RIP assay. (C) Subcellular levels of HBV mRNA in cytoplasm and nucleus were determined by qPCR (% of 1.2 HBV). (D) Cytoplasmic and nuclear HBV RNAs levels were evaluated by RNAscope assay. (E-H) 1.2 HBV or 1.2 HBV ΔHBc construct was transfected into Huh7 cells with ELAVL1 knockdown. (E) The levels of ELAVL1 and HBc were determined by WB. (F) Secreted HBeAg and HBsAg levels were determined by ELISA (% of 1.2 HBV). (G) Subcellular levels of HBV mRNA in cytoplasm and nucleus were determined by qPCR (% of 1.2 HBV).

replication in HepAD38 and HBV-infected HepG2-NTCP cells (S7 Fig). This suggests ELAVL1 is the specific adaptor for both the nucleocytoplasmic transport and stability of HBV RNAs, rather than NXF3 or EIF4E.

The interaction of ELAVL1 and CRM1 is mediated by protein ligands ANP32A and ANP32B which contain leucine-rich nuclear export signals [31]. To test whether ANP32A and ANP32B also bridge HBV RNA-ELAVL1 complex and CRM1, we knocked down these two proteins using siRNA (Fig 7A). Deficiency of either ANP32A or ANP32B suppressed secreted HBV DNA, antigen levels (Fig 7B), and HBV RNAs export (Fig 7C). Likewise, RIP assay revealed that ANP32A and ANP32B inhibition strongly disrupted CRM1-HBV RNAs interaction (Fig 7D) but not ELAVL1-HBV RNAs interaction (Fig 7E). All these data suggested that HBV RNAs were recognized by ELAVL1, which in turn binds to ANP32A and ANP32B and subsequently recruited CRM1 (Fig 7F).

## ELAVL1 protects nuclear HBV RNAs from DIS3⁺RRP6⁺ RNA exosome degradation

Our data showed that impaired viral RNA export did not result in accumulation of nuclear HBV RNA, suggesting a balance between RNA export and degradation. To confirm this hypothesis and explore how HBV RNAs were degraded, we performed an siRNA screen of known nuclear RNases [32]. DIS3, EXOSC4, RRP4, or RRP6 were sufficiently knocked down

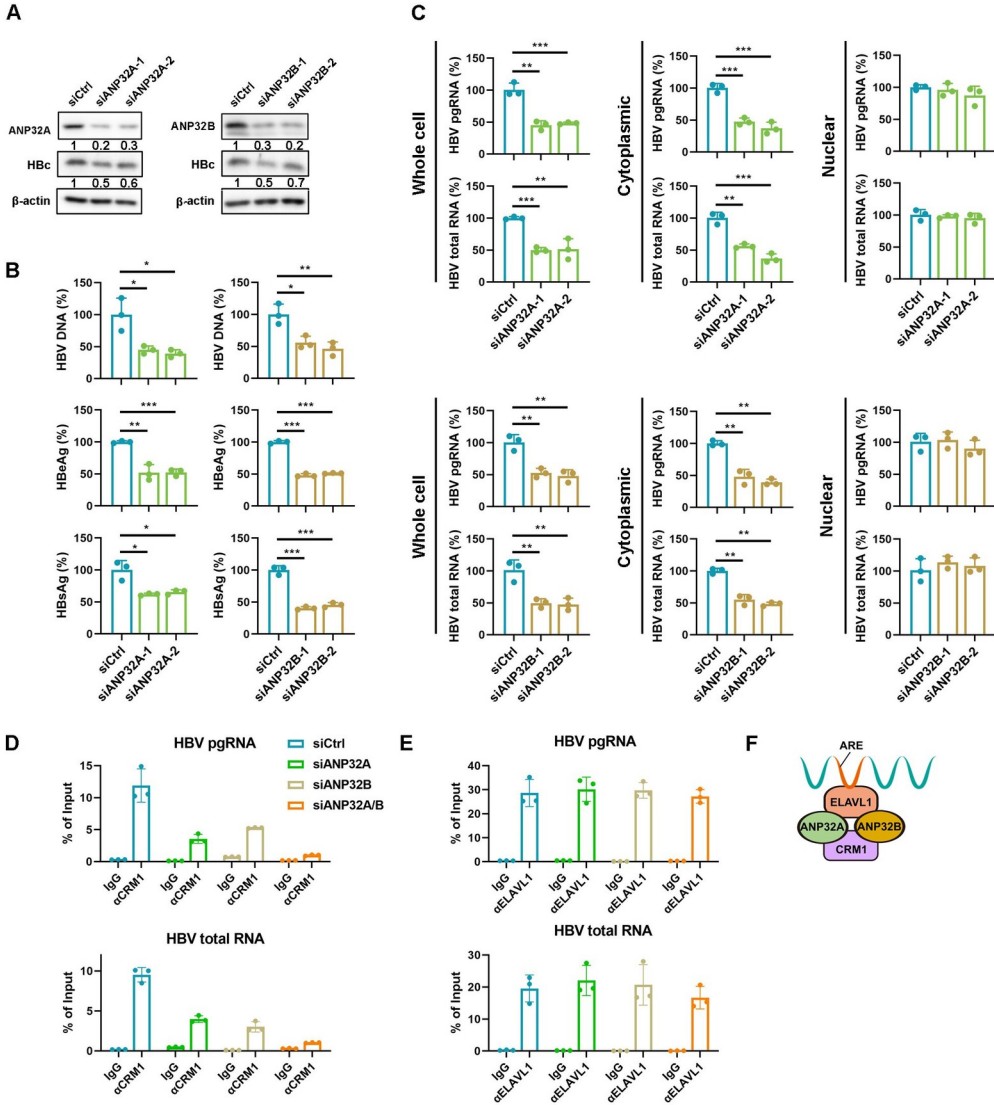

**Fig 7. ANP32A and ANP32B bridge HBV RNA-ELAVL1-CRM1 and mediate HBV RNAs nuclear export.** (A-D) HepG2-NTCP cells were transfected with ANP32A or ANP32B targeted siRNA. The cells were infected with HBV at an MOI = 200 and were maintained with DMEM medium supplemented with 2.5% DMSO for 5 days. (A) Levels of ANP32A, ANP32B, and HBc were determined by WB. (B) Levels of HBV DNA in supernatants were determined by qPCR (% of siCtrl). Secreted HBeAg and HBsAg levels were determined by ELISA (% of siCtrl). (C) Subcellular levels of HBV mRNA in cytoplasm and nucleus were determined by qPCR (% of siCtrl). Graphs were shown as mean ± SD. *, p < 0.05; **, P < 0.01; ***, p < 0.001. (D and E) ANP32A and (or) ANP32B knockdown HepAD38 cells were maintained in DMEM medium containing 2% DMSO for 2 days. The binding of (D) HBV RNA-CRM1 and (E) HBV RNA-ELAVL1 were detected by RIP assay. (F) Schematic description of the HBV RNA binding proteins complex.

in HepAD38 cells (Fig 8A). Knockdown of DIS3 and RRP6 elevated supernatant HBV DNA, secreted antigen levels (Fig 8B), HBV RNAs levels, and HBc levels (Fig 8C and 8D). Since DIS3 and RRP6 are two catalytic subunits of the RNA exosome [33], these data indicate the RNA exosome degrades HBV RNA. To further confirm this, we used HBV-infected HepG2-NTCP cells. Similarly, DIS3 knockdown increased supernatant HBV DNA, secreted antigens, and HBc (Fig 8E and 8F). Fraction analysis revealed both cytoplasmic and nuclear HBV RNAs levels increased upon DIS3 knockdown (Fig 8G). We next investigated whether HBV RNA

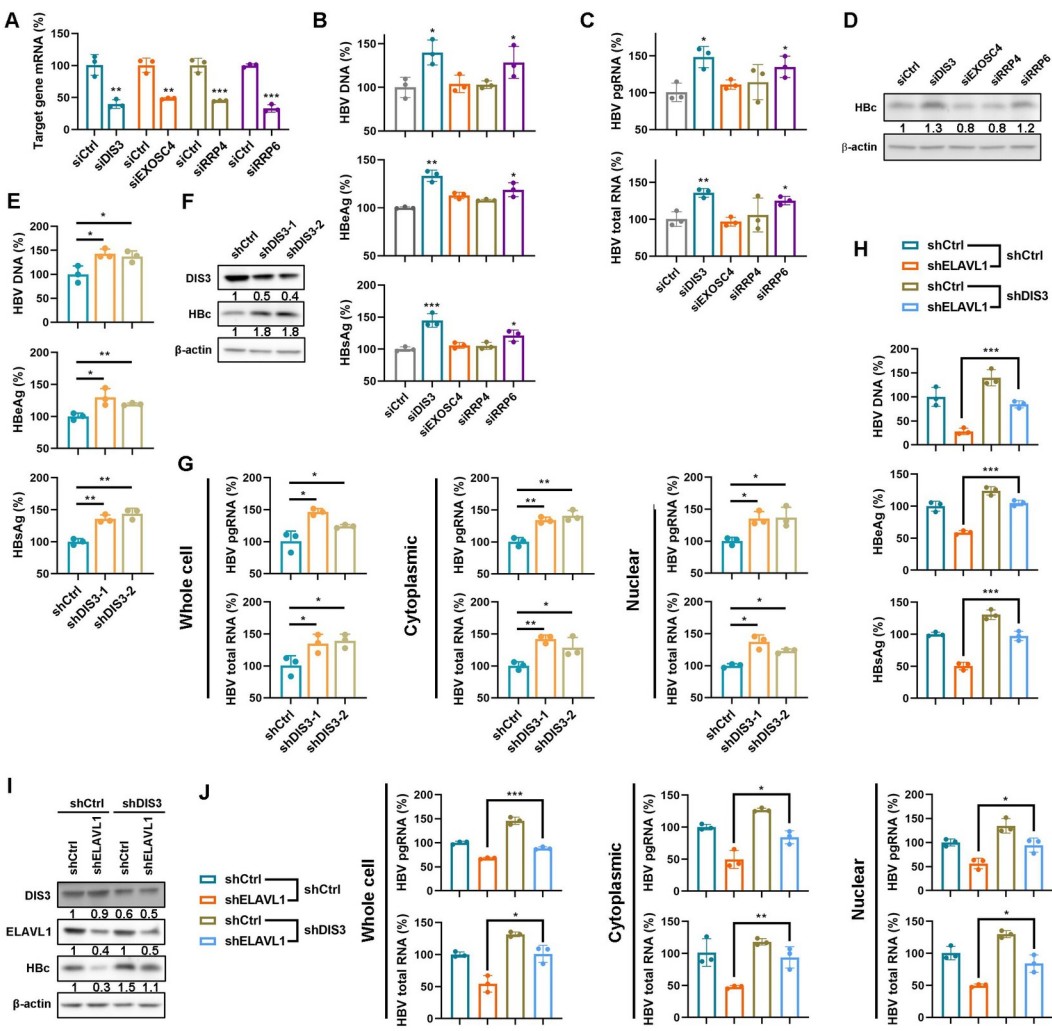

**Fig 8. ELAVL1 protects nuclear HBV RNAs from DIS3+RRP6+ RNA exosome degradation.** (A-D) HepAD38 cells were transfected with DIS3, EXOSC4, RRP4, and RRP6 targeted siRNA respectively and were maintained with DMEM containing 2% DMSO for 2 days. (A) Knockdown efficiency of every siRNA was detected by qPCR (% of siCtrl). (B) Levels of HBV-DNA in supernatants were determined by qPCR (% of siCtrl). Levels of secreted HBeAg and HBsAg were determined by ELISA (% of siCtrl). (C) Levels of intracellular HBV pgRNA and total RNA were determined by qPCR (% of siCtrl). (D) Levels of HBc were determined by WB. (E-G) HepG2-NTCP cells infected with lentivirus expressing DIS3-targeted shRNAs were pretreated with 2.5% DMSO for 2 days following HBV infection at an MOI of 200 and were maintained with DMEM containing 2.5% DMSO for 7 days. (E) Levels of HBV DNA in supernatants were determined by qPCR (% of shCtrl). Levels of secreted HBeAg and HBsAg were determined by ELISA (% of shCtrl). (F) Levels of DIS3 and HBc were determined by WB. (G) Subcellular levels of HBV mRNA in cytoplasm and nucleus were determined by qPCR. (H-J) HepG2-NTCP cells with DIS3 knockdown were infected with lentivirus expressing ELAVL1-targeted shRNAs and were pretreated with 2.5% DMSO for 2 days following HBV infection at an MOI of 200 and maintained with 2.5% DMSO for 7 days. (H) Levels of HBV DNA in supernatants were determined by qPCR (% of shCtrl). Levels of secreted HBeAg and HBsAg were determined by ELISA (% of shCtrl). (I) Levels of DIS3, ELAVL1 and HBc were determined by WB. (J) Subcellular levels of HBV mRNA in cytoplasm and nucleus were determined by qPCR (% of shCtrl). Graphs were shown as mean ± SD. *, p < 0.05; **, p < 0.01; ***, p < 0.001.

reduction upon ELAVL1 knockdown was related to the RNA exosome. ELAVL1-targeted shRNA was introduced into HBV-infected HepG2-NTCP cells with stable DIS3 knockdown. Interestingly, DIS3 inhibition rescued HBV replication, as shown by different HBV markers (Fig 8H and 8I). Notably, both cytoplasmic and nuclear HBV RNAs increased with DIS3 shRNA even upon ELAVL1 or CRM1 knockdown (Fig 8J and S8). Collectively, these data

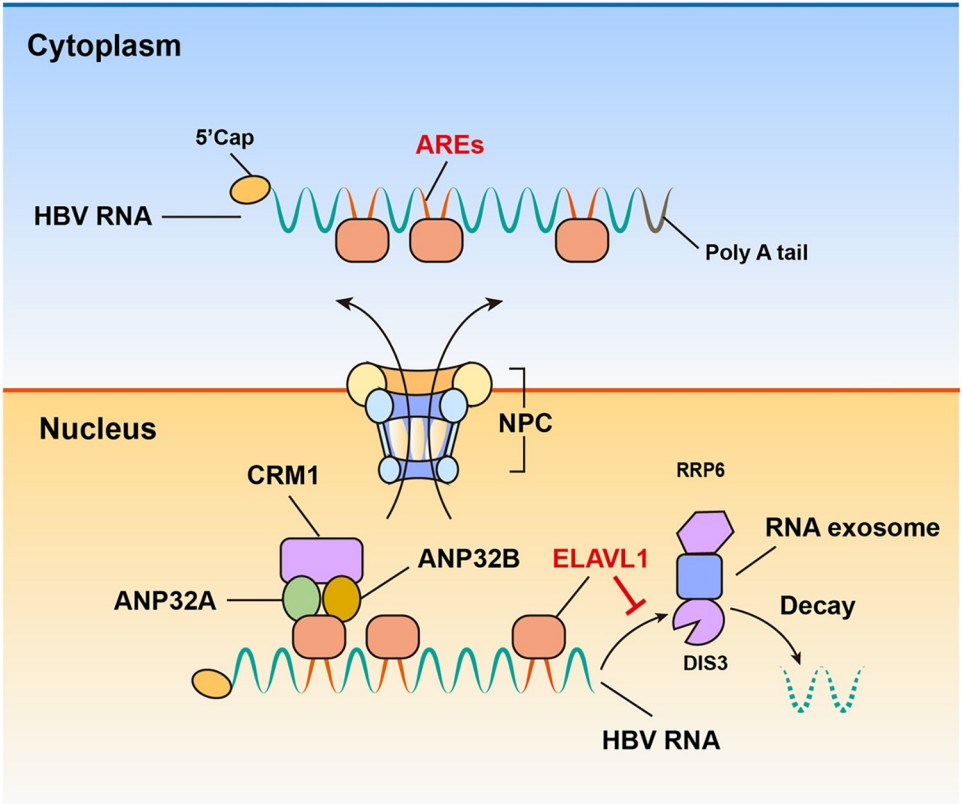

**Fig 9. Graphic summary of the key findings.** ELAVL1 escorts HBV RNAs from nucleus to cytoplasm via CRM1 pathway. ELAVL1 binds to HBV RNAs via AU-rich elements in the viral RNAs which in turn associated with ANP32A and ANP32B and subsequently recruits CRM1 to export from nucleus. By binding to HBV RNAs, ELAVL1 also protects the viral RNAs against DI3+RRP6+ RNA exosome degradation.

indicate the $DIS3^+RRP6^+$ RNA exosome degrades nuclear HBV RNAs, and ELAVL1 protects HBV RNAs from degradation.

In sum, we concluded that ELAVL1 escorts HBV RNAs from nucleus to cytoplasm via CRM1 pathway. ELAVL1 binds to HBV RNAs via AU-rich elements in the viral RNAs which in turn associated with ANP32A and ANP32B and subsequently recruits CRM1 to export from nucleus. By binding to HBV RNAs, ELAVL1 also protects the viral RNAs against $DI3^+RRP6^+$ RNA exosome degradation (Fig 9).

## Discussion

Previous studies regarding mechanisms of HBV RNAs nuclear export remain elusive. It has been reported that the HBV PRE can enhance the expression of HBV transcripts and promote the nuclear export of unspliced HBV RNAs [14,34–37]. In contrast, the PRE is not involved in the nuclear export of the full-length pgRNA that utilizes TAP/NXF1 dependent pathway [16]. In addition, there are evidences show that PRE contribute to pgRNA stability, but has little effect on its nuclear export [17]. Of note, these studies were mainly conducted in plasmid transfection models, and not validated in HBV infection models. Through a Genome-wide CRISPR Screen, Hyrina et al. uncovered ZCCHC14, together with TENT4A/B, interact with HBV PRE region and stabilize HBV mRNA by promoting 3′ end tailing, and thereby likely preventing degradation by exonucleases [38]. Here, we demonstrated that AREs are required

for HBV RNA nuclear export via ELAVL1-CRM1 pathway. Together, these data suggesting different and distinct roles of PRE and ARE in HBV RNA post-transcriptional regulation.

ELAVL1 is a member of ELAVL family of RNA-binding proteins. It contains three RNA recognition motifs and selectively binds to AREs in the 3' untranslated regions (3'UTR) of mRNAs. By binding to AREs, ELAVL1 plays a role in stabilizing ARE-containing mRNAs [39,40]. Besides, ELAVL1 is also an adaptor mediating mRNA containing AREs nucleoplasm transport via CRM1 [31,41]. It plays a physiological role in mediating the cellular response to apoptotic, proliferating and pathogens [42]. Research has demonstrated that ELAVL1 plays a regulatory role in various cell-cycle related proteins, potentially influencing HBV replication [43]. In our cell culture experiments, we aimed to circumvent the influence of ELAVL1 on the cell cycle. To achieve this, we consistently applied DMSO treatment in the cell culture medium to induce cell cycle arrest. Consequently, the observed effects in our experiments are unlikely to be associated with cell cycle-related factors.

ELAVL1 is also a known host factor of many viruses. Hepatitis C virus replication is regulated by ELAVL1, through the interaction with 3' UTR of the viral RNA and PLK1-ELAVL1--miR-122 signaling [44–46]. ELAVL1 plays an important role in adenovirus replication, by up-regulating viral IVa2 mRNA expression and stability [47]. Sindbis virus hijacks ELAVL1 to stabilize its transcripts and promote productive infections [48]. ELAVL1 is proposed to interacts with human immunodeficiency virus type 1 (HIV-1) reverse transcriptase, and modulates reverse transcription [49]. However, further study demonstrates that ELAVL1 does not affect the kinetics of HIV-1 reverse transcription *in vitro*, even on RNA templates that contain AREs [50]. For influenza virus, ELAVL1 contribute to control of viral mRNA splicing [51]. Interestingly, ELAVL1 exhibits an inhibitory effect in ZIKV replication [52]. Likewise, ELAVL1 suppresses Coxsackievirus B3 replication by displacing proviral factor PCBP-2 at the cloverleaf RNA [53]. As positive-strand RNA virus must balance the availability of its genomic template for various viral processes at different stages of its life cycle, the limitation of virus replication mediated by ELAVL1 can coordinate the availability of genomic RNA templates for translation, replication, or encapsidation [53]. A recent study reported that ELAVL1 knockdown decreased the levels of HBV RNAs in HepAD38 cells without detailed mechanism exploration [43]. In this study, we dissected the molecular mechanisms involved in HBV RNA export. ELAVL1 binds to HBV mRNAs in the nucleus and then escorts them to the cytoplasm and protects them from RNA exosome degradation.

For most nuclear mRNAs, the final destiny is either exported to the cytoplasm or degraded in the nucleus. The RNA exosome is a multiprotein complex that possesses RNase activity and is a key player in processing, quality control and degradation of RNAs [33]. The nuclear RNA exosome consists of a barrel-shaped nine-subunit core (EXO9) and two catalytic subunits. While EXOSC10 binds to the cap side of EXO9, DIS3 interacts with EXO9 at the bottom. RNA substrates are threaded through a central channel to the catalytic core of DIS3 for degradation [33]. In this study, we observed that DIS3 knockdown rescued the intracellular HBV RNA levels when their nuclear export pathway was hindered. Although the mechanisms by which the exosome targets HBV RNAs remain unknown, our data suggest an important role of RNA exosome in nuclear HBV RNA degradation.

The export of both protein-coding and non-coding RNA molecules from the nucleus into the cytoplasm is essential for gene expression. This requires the constant transport of RNAs of diverse sizes, structures, and functions through the intricate nuclear pore complexes (NPCs) via export receptors and adaptor proteins. While a large number of mRNAs utilize the TREX, TREX-2 and NXF1 receptors to traverse the NPCs, a subset of mRNAs employs CRM1, the primary export receptor for proteins, instead [54]. CRM1 itself does not directly associate with RNA. Rather, it is recruited to RNAs through interaction with adaptor proteins harboring a

nuclear export signal (NES) that bind either to the RNA directly or to other RNA-binding proteins. The mechanism by which CRM1 facilitates RNA export is analogous to how it exports proteins—it recognizes NES-containing adaptor proteins that in turn bind to the cargo, which in this case is either the RNA itself or an RNA-binding protein complex [24]. We demonstrate here that HBV RNAs associate with ELAVL1, which in turn binds to ANP32A and ANP32B, and these interactions subsequently recruit CRM1. However, as HBx mRNA contains no ARE, how it gets exported need further investigations. Besides, the current work was based on genotype D only, and the relative contribution of ELAVL1-CRM1 in other HBV genotypes remain to be tested experimentally.

Many of the ELAVL1 target mRNAs encode proteins important for cell growth, tumorigenesis, angiogenesis, tumor inflammation, invasion and metastasis [42]. ELAVL1 overexpression or overactivation is known to correlate with high-grade malignancy, drug resistance and poor prognosis in many cancers [55]. Thus, ELAVL1 has emerged as an attractive drug target for cancer therapy. Strategies including inhibiting ELAVL1 translocation from the nucleus to the cytoplasm, inhibiting the ability of ELAVL1 to bind target RNA and silencing ELAVL1 expression are proposed [55]. In this study, we reported that HBV RNA nuclear export is a druggable antiviral target. CMLD-2, an inhibitor of ELAVL1 competitively binds ELAVL1 disrupting its interaction with ARE-containing mRNAs, restricts HBV production, inhibits HBV RNA nuclear export, and reduced viral RNA stability. Although the inhibitory effect is not dramatic due to CMLD-2's limited potency in ELAVL1-HBV RNA binding impairement, our study provides a proof of concept illustrating that targeting ELAVL1 holds promise as a potential approach to suppress HBV. The potential usage of ELAVL1 inhibitor as an add-on therapy to control HBV infection should be evaluated.

## Materials and methods

### Ethics statement

Animals used in this study were treated in accordance with the guidelines on humane care, and the protocols were approved by the Ethic Committee of Animal Facility, Wuhan University.

### Cell lines

HepAD38 cells, HepG2-NTCP cells, Huh7 cells, and HEK293T cells were grown in Dulbecco's modified Eagle's medium (DMEM, Bio-channel, Cat #: BC-M-005, China) supplemented with 10% fetal bovine serum (FBS, LONSERA, Cat #: S711-001S, Uruguay), 100 U/ml penicillin and 100 μg/ml streptomycin (GIBCO, Cat #: 15140–122, China). All the four cell lines were cultured at 37˚C in a 5% $CO_2$ incubator. Primary human hepatocytes were purchased from Liver Biotechnology (Shenzhen) Corporation.

### RT–qPCR

RNA was extracted using a commercial RNA extraction kit following the manufacturer's protocol (S1 Table). cDNA was synthesized using the TOYOBO ReverTra Ace qPCR RT Master Mix (S1 Table) with gDNA Remover kit following the manufacturer's instruction. qPCR was performed using the ROCH FastSmart Essential DNA Green Master mix (S1 Table) on Light-Cycler 480 II with qPCR assays. Primers used for RT–qPCR are listed in S3 Table. Of note, the HBV RNA fragments to be amplified are 2112–2278 for pgRNA and 1551–1778 for total RNA.

## Western blotting

Total protein was extracted from cells or livers with RIPA buffer supplemented with protease inhibitor cocktail. Protein lysates were separated via SDS-PAGE and transferred onto PVDF membranes. The membranes were blocked in 5% skim milk for 1 h and incubated with the indicated primary antibodies (S2 Table) and then secondary antibodies conjugated HRP. Protein bands were detected using GeneGnome XRQ chemiluminescence imaging system (XRQ-NPC, GeneGnome, Hong Kong, China).

## Animal studies

For AAV-sh*Elavl1* transduction, 6–8 weeks old C57BL/6J mice were purchased from China Three Gorges University Laboratory Animal Center. Mice were injected with the recombinant AAV-HBV1.2 and AAV-sh*Elavl1* ($1 \times 10^{11}$ and $5 \times 10^{11}$ vg diluted in 200 μL PBS respectively) through tail vein. The blood samples and liver tissues were collected at indicated time points. The serum samples were obtained by centrifugation at 3500 rpm for 15 min at 4˚C. Serological HBsAg and HBeAg along with intrahepatic viral RNA and protein were determined.

## Nuclear and cytoplasmic RNA extraction

For adherent cells, samples were harvested by trypsin-EDTA and then were collected by centrifugation at 500 rpm for 5 min. The cell pellets were washed twice with precooled PBS. The supernatant was carefully removed leaving the cell pellet as dry as possible. The samples were fractionated by a commercial nuclear and cytoplasmic extraction reagents kit (S1 Table) following the instruction from manufacturer.

For mouse tissue, 60 mg sample was cut into small pieces and were washed with precooled PBS twice. The supernatant of the samples was removed leaving sediment as dry as possible. The tissues were homogenized using a homogenizer in the appropriate volume of cytoplasmic protein extraction reagent A provide by the commercial kit above. Then the samples were subjected to cytoplasmic and nuclear fractionation as manufacture's instruction.

## Cell transfection

For DNA transfection, cells were transfected with PEI MAX 40K Reagent (S4 Table) according to the manufacturer's recommendations.

For siRNA transfection, cells were transfected 6 hours with Lipofectamine RNAiMAX (S4 Table) following instructions from manufacturer.

## Virus production and infection

For the production of pseudotyped lentiviral supernatant, on the day before transduction, HEK293T cells were seeded at T25 culture flask. 12 hours after, the pseudoviruses were generated by co-transfecting HEK293T cells with pLKO.1-shRNA plasmids (2 μg), pMD.2G envelope plasmid (0.5μg), and a packaging vector psPAX2 (1.5 μg) using the PEI MAX 40K Reagent following the manufacturer's recommendations. Supernatants were harvested after 48 h and 72 h and then were filtered with a 0.45-μm membrane to remove cell debris.

For lentivirus infection, cells were seeded 12 hours before infection. Cell culture medium was supplemented with proper dose of pseudotyped lentiviral supernatant containing 8 μg/ml polybrene (S4 Table). Infection culture medium was replaced by complete culture medium after 24 hours.

For the production of adeno-associated virus supernatant, HEK293T cells were seeded into 15 cm dishes 12 hours before transfection. Virus supernatant was generated by co-transfecting

HEK293T cells with Px552-shRNA, pAAV2/8RC, and pHelper at 1:1:1 molar ratio with the PEI MAX 40K Reagent (S4 Table) following the manufacturer's recommendations. Supernatants were collected at 72 hours after transfection and were filtered through a 0.22-μm membrane to remove cell debris. Then the supernatants were concentrated with Cytiva protein concentration unit (Cytiva, Cat #: 28932363, United States). Titers of the recombinant AAV was determined by qPCR.

For HBV production, HBV stocks were obtained from the culture supernatants of HepAD38 cells and was concentrated by a centrifugal filter (Millipore, Cat #: UFC910096, Ireland). Titers were determined by an HBV DNA diagnostic kit (S1 Table).

For HBV infection, HepG2-NTCP cells were incubated with HBV at 200 MOI in medium containing 4% PEG 8000 and 2.5% DMSO for 24 hours. The viral mixture was discarded and the cells were washed quintic with PBS each and then refreshed with complete medium containing 2.5% DMSO medium for further incubation. Supernatants were collected at indicated time points.

## Enzyme linked immunosorbent assay

Secretions of HBsAg and HBeAg in the culture supernatants were detected by ELISA kits according to the manufacturers' protocols. The ELISA kits used in this study were purchased from Shanghai Kehua Bio-Engineering (S1 Table). The absorbance was determined using a microtiter plate reader at 450 nm.

## Stable knockdown cell lines construction

Pseudotyped lentiviral supernatant expressing ELAVL1, CRM1, or DIS3 targeted shRNA were produced as describe in the section *Virus production and infection*. Cells were seeded in 6-well culture plate. The cells were infected with the lentivirus in culture media containing 8 μg/mL polybrene. The target cells were selected with 2 μg/ml puromycin 24 hours post infection. Culture medium supplemented with 2 μg/ml puromycin was refreshed every other day. The knockdown efficiency of target gene was confirmed by WB assay or qPCR assay.

## HBc-associated DNA extraction

The tissue homogenate, prepared using liver tissue and lysed with a buffer containing 50 mM Tris-HCl, 50 mM NaCl, 1 mM EDTA, and 1% NP-40 at pH 7.4, was subjected to centrifugation at 14,000 rpm for 3 minutes at 4˚C. The supernatant was then transferred to a new 2 ml EP tube. Subsequently, 1M $MgCl_2$ and 8 μl of 10 mg/ml DNase I were added, and the mixture was incubated at 37˚C for 30 minutes. Following this, 40 μl of 0.5 M EDTA (pH 8.0), 20 μl of 20 mg/ml proteinase K, and 80 μl of 10% SDS were added, and the incubation continued at 55˚C in a water bath for 2 hours. Next, 500 μl of Tris-saturated phenol and chloroform were added, and the samples were incubated at room temperature for 2–3 minutes before being centrifuged at 12,000 rpm for 5 minutes at room temperature. The upper layer was carefully transferred to a new 2 ml EP tube. Then, 0.7 volumes of isopropanol, 0.1 volume of 3 M NaAc (pH 5.2), and 15 μg of Yeast RNA were added, and the mixture was precipitated overnight at -20˚C. The resulting pellet was collected by centrifugation at 14,000 rpm for 15 minutes at 4˚C and washed with 70% ethanol before being air-dried. Subsequently, the pellet was dissolved in 20 μl of RNase water, heated at 70˚C for 10 minutes, and the HBV core-associated DNA solution was obtained.

## HBV promoter/enhancer activity assay

The activity of HBV promoters/enhancers were evaluated using a commercial dual-luciferase reporter assay kit from Promega according to the manufacturers' protocols, detailed information of the kit was listed in S1 Table.

## Protein expression and purification

pET28a-6xHis-ELAVL1 was expressed in BL21 chemically competent cells. The cells were grown at 37˚C to an OD600 of 0.6–0.8, and then protein expression was induced by the addition of 1 mM isopropyl β- d-1-thiogalactopyranosideand were cultured at 16˚C for another 16 h. Then the cells were pelleted and resuspended in nondenaturing lysis buffer (250 mM $NaH_2PO_4$, 300 mM NaCl, pH = 8.0) supplemented with protease inhibitor PMSF and lysed in ice for 30 min. The cells are ultrasonically lysed on ice. The cell lysates were centrifuged at 10,000 g for 30 minutes. Supernatant was incubated with an Ni- His-tag Purification resin (Beyotime, Cat #: P2218, China) for 2 h at 4˚C, and then the resins were passed through via gravity flow. The resins were washed 5 times with 2-bed volumes of non-denaturing washing solution (50 mM $NaH_2PO_4$, 300 mM NaCl, 2 mM imidazole, pH 8.0), followed by 1-bed volumes of elution buffer (50 mM $NaH_2PO_4$, 300 mM NaCl, 50 mM imidazole, pH 8.0) for 5 times. The eluted protein solutions were concentrated in a 30-kDa molecular weight cutoff Cytiva centrifugal filtration device (Cytiva, Cat #: 28932363, United States).

## RNA pulldown

The pgRNA and its mutant were synthesized by an *in vitro* transcription kit (S1 Table) following manufacturer's instructions. Linearized pCDNA3.1-pgRNA plasmid was used as template. The 3´ terminus desthiobiotinylated pgRNA were synthesized from the linearized DNA templates with T7 RNA polymerase, GTP, ATP, CTP, and UTP. The pgRNA was treated with DNase I for 20 min and then phenol–chloroform extracted. The RNA was further purified using a commercial kit (S1 Table).

The streptavidin-coated magnetic beads were washed by an alkaline solution containing 0.1 M NaOH and 0.05 M NaCl for 2 min in room temperature in a rotator. The beads were washed twice and were resuspended with 0.1 M NaCl solution. Following a 2 min rotating, beads pellets were collected. The pellets were resuspended with 2 × washing and binding buffer (W&B buffer) and equal volume biotiylated pgRNA was added. The mixtures were incubated at room temperature for 1 hour and beads were collected on magnet. The beads were washed twice with W&B buffer (5 mM Tris-HCl (pH 7.4), 1M NaCl). Cell lysates were mixed with the beads for 3 hours at 4˚C with rotation. The beads were collected by centrifugation and were eluted with 2 mM biotin in PBS for 3 hours in a rotator. The eluates were subjected to WB or LC-MS/MS assays.

## LC-MS/MS

The proteins captured by RNA pulldown (described in the section *RNA pulldown*) were digested in urea buffer as previously described [56]. The obtained peptides were desalted with the C18 stage tips (Thermo Fisher Scientific) and analyzed by an Orbitrap Exploris 480 mass spectrometer with the FAIMS Pro interface. Sample analysis protocol was as previously described [57]. Raw data are available on ProteomeXchange Consortium (reference number: 1-20230724-064428-2501911; project accession ID: PXD044015).

## RNA electrophoretic mobility shift assay (REMSA) and super shift REMSA

6×His tagged ELAVL1 was expressed and purified *in vitro* as describing in the section Protein expression and purification. The pgRNA probe (5'-cuuuuccuaauauacauuuacaccaagacauuau-caaaaaaugugaacaguuuguaggcccacucacaguuaaugagagaaaagaagauugcaauugauuaugccugccag-guuuuauccaaagguuaccaaauauuuaccauuggauaagggguauuaaaccuuauuauccagaacaucuaguuaau-cauuacuuccaaacuagacacuauuuacacacucuauggaag-3') or its mutant (5'-cuuuuccuaauauacaucuccaccaagacauuaucaaaaaaugugaacaguuuguaggcccacucacaguuaauga-gaaaagaagauugcaauugauuaugccugccaggguuuuauccaaagguuaccaaauauucucccauuggauaagg-guauuaaaccuuauuauccagaacaucuaguuaaucauuacuuccaaacuagacacuaucuccacacucuauggaag-3') were synthesized by a T7 mScript Standard mRNA Production System kit (S1 Table) following manufacturer instruction. Briefly, pCDNA3.1-T7-pgRNA plasmid was linearized by Pme I and was purified by a commercial DNA purification kit (Vazyme, Cat. No.: DC301-01). The linear DNA template along with T7 RNA polymerase and GTP, ATP, CTP, UTP were incubated in reaction buffer for 20 min. Probes were treated with DNase I for 20 min and then were purified by a commercial RNA purification kit (S1 Table) following phenol-chloroform extraction.

The 3' terminus of the RNA probes were labeled with desthiobiotin by using a commercial RNA 3' end desthiobiotinylation kit (S1 Table) following manufacturer instruction. Briefly, the purified RNA prepared before was incubated with T4 RNA ligase and biotinylated cytidine bisphosphate in ligase reaction buffer supplemented with 15% PEG at 16°C overnight. The labeled RNA probes were extracted with phenol-chloroform and purified using a commercial kit (S1 Table).

The purified His-ELAVL1 and the prepared pgRNA probe were incubated in binding buffer (10 mM Tris (pH 7.5), 1 mM EDTA, 100 mM KCl, 100 μM DTT, 5% vol/vol glycerol, 10 μg/ml BSA) at 37°C for 5 min. For cool competent REMSA, competent probes (unlabeled probes) were incubated with His-ELAVL1 for 15min at room temperature before incubation with desthiobiotin labeled probes. For super-shift REMSA, before the incubation of the labeled probes at 37°C for 5 min, competent probes, His-ELAVL1, and ELAVL1-targeted antibody were incubated in binding buffer for 15 min at 37°C.

The incubated mixtures (9 μl) and 1 μl 10× glycerol stock solution (10 mM Tris, 1 mM EDTA, 50% vol/vol glycerol, 0.001% wt/vol bromophenol blue) were mixed together and immediately subjected to electrophoresis. The samples were loaded into a 0.5% agarose gel and were electrophoresed at 120 v for half an hour in 1% TBE buffer.

The RNA along with proteins were transferred to uncharged nylon membranes with 0.5% TBE buffer. The nylon membranes were irradiated at 254 nm for 1 min 45 s at 1.5 J/cm$^2$ and were incubated with biotin-targeted antibody linked with HRP for 1 hour at room temperature. Ultimately, the membranes were exposed by GeneGnome XRQ chemiluminescence imaging system (XRQ-NPC, GeneGnome, Hong Kong, China).

## RNA immunoprecipitation

Estimated $1 \times 10^7$ cells were collected per RIP. the cells were pelleted by centrifugation 1, 000 g for 10 min at 4°C and washed twice with 10 ml of pre-cooled phosphate buffered saline (PBS). Final cell pellets were resuspended with an approximately equal volume of polysome lysis buffer (100 mM KCl, 5 mM MgCl$_2$, 10 mM HEPES (pH 7.0), 0.5% NP40, 1 mM DTT, 400 μM VRC, RNase-DNase-free H$_2$O) supplemented with RNase inhibitors (50 U/ml) and protease inhibitors cocktail. Clumps of cells were broken down by pipetting up and down several times and were centrifuged at 15,000 g for 15 min to clear lysate of large particles following an incubation on ice for 10 min.

Protein A/G-coated agarose beads (70 µl) were resuspended in NT2 buffer (50 mM Tris-HCl (pH 7.4), 150 mM NaCl, 1 mM MgCl2, 0.05% NP40, RNase-DNase-free $H_2O$) supplemented with 5% BSA and were rest for 1 hour at room temperature. Antibody was added to bead slurry and were incubated for 4 hours tumbling end over end at 4˚C. Immediately before use, wash antibody-coated beads with 1 ml of ice-cold NT2 buffer 4 times. After the final wash, resuspend beads in 850 µl of ice-cold NT2 buffer.

The cleared lysate (100µl) was added to the resuspended beads and were incubated for 4 h at 4˚C tumbling end over end. The beads were pelleted and were washed 4 times with 1 ml of ice-cold NT2 buffer. The beads were Resuspended in 100 µl of NT2 buffer. The RNA was released from the RNP components and were isolate by Trizol reagent.

## RNAscope

Samples were prepared by a RNAscope sample preparation kit (S1 Table) following the manufacturer recommendation. Subcellular distribution and levels were detected by a commercial RNAscope detection kit (S1 Table) according to the instruction form manufacturer. The confocal laser scanning microscope we used to detect signals in cell was NIKON Eclipse Ti (software: Eclipse C2). The fluorescent channels we used were DAPI (EX 405 nm, EM 417–477 nm); probe dye 570 (EX 561 nm, EM 570–690 nm). The confocal laser scanning microscope we used to detect signals in tissue was Leica TCS SP8 (software: LAS X Version 3.4.2.18368). The fluorescent channels we used were DAPI (EX 405 nm, EM 417–480 nm); probe dye 570 (EX 587 nm, EM 570–690 nm).

## Quantification and statistical analysis

Data were presented as mean ± SD or mean ± SEM as indicated. Comparison between two groups was performed by one-way ANOVA. Statistical analysis was conducted with GraphPad Prism 8.0. $^*p < 0.05$; $^{**}p < 0.01$; $^{***}p < 0.001$.

## Supporting information

**S1 Fig. ELAVL1 is a potential factor regulating HBV replication.** (A) Peptide count of the pgRNA binders identified by the RNA-pulldown-LC-MS/MS. (B) Gene ontology analysis (cluster analysis) for the pgRNA binders' network. Red node indicates the cluster of nucleocytoplasmic transport regulators.
(TIF)

**S2 Fig. ELAVL1 mediates HBV RNAs post-transcriptional regulation.** (A-C) ELAVL1 knockdown HepG2-NTCP cells were pretreated with 2.5% DMSO for 2 days following HBV infection at an MOI of 200 and were maintained with DMEM containing 2.5% DMSO for 7 days. (D-F) ELAVL1 knockdown HepAD38 cells were maintained in DMEM medium containing 2% DMSO for 2 days. (A and D) Knockdown efficiency was confirmed by WB. (B and E) The levels of HBV DNA in culture supernatant were determined by qPCR (% of shCtrl). The levels of HBeAg and HBsAg in culture supernatant were determined by ELISA (% of shCtrl). (C and F) Subcellular levels of HBV RNAs in cytoplasm and nucleus were determined by qPCR (% of shCtrl). Graphs show mean ± SD. $^*p < 0.05$; $^{**}p < 0.01$; $^{***}p < 0.001$.
(TIF)

**S3 Fig. ELAVL1 deficiency restrains HBV replication.** (A-C) The lentivirus expressing ELAVL1 infected HepG2-NTCP cells with stable ELAVL1 knockdown. The cells were pretreated with 2.5% DMSO for 2 days following HBV infection at an MOI of 200 and were maintained with DMEM containing 2.5% DMSO for 7 days. (A) Levels of ELAVL1 and HBc were

evaluated by WB. (B) The levels of HBV DNA in culture supernatant were determined by qPCR (% of shCtrl). The levels of HBeAg and HBsAg in culture supernatant were determined by ELISA (% of shCtrl). (C) Subcellular levels of HBV RNAs in cytoplasm and nucleus were determined by qPCR (% of shCtrl). (D-G) HepAD38 cells were treated with CMLD-2 as indicated dose for 48 hours. (D) Levels of HBV-DNA in supernatants were determined by qPCR (% of NC). Levels of secreted HBeAg and HBsAg were determined by ELISA (% of NC). (E) Levels of HBc were determined by WB. (F) Subcellular levels of HBV RNAs in cytoplasm and nucleus were determined by qPCR (% of NC). (G) Cell viability was evaluated by CCK-8 assay. Graphs show mean ± SD. *p < 0.05; **p < 0.01; ***p < 0.001.
(TIF)

**S4 Fig. Homology analysis for AREs in different HBV genotypes.**
(TIF)

**S5 Fig. CRM1 regulates HBV RNAs nuclear export.** (A-C) CRM1 knockdown HepG2-NTCP cells were pretreated with 2.5% DMSO for 2 days following HBV infection at an MOI of 200 and were maintained with DMEM containing 2.5% DMSO for 7 days. (D-F) CRM1 knockdown HepAD38 cells were maintained in DMEM medium containing 2% DMSO for 2 days. (A and D) Knockdown efficiency were confirmed by WB. (B and E) The levels of HBV DNA in culture supernatant were determined by qPCR (% of shCtrl). The levels of HBeAg and HBsAg in culture supernatant were determined by ELISA (% of shCtrl). (C and F) Subcellular levels of HBV RNAs in cytoplasm and nucleus were determined by qPCR (% of shCtrl). Graphs were shown as mean ± SD. *, p < 0.05; **, p < 0.01; ***, p < 0.001.
(TIF)

**S6 Fig. NXF1 regulates ELAVL1 and CRM1 expression to support HBV RNAs nucleocytoplasmic transport.** (A-E) NXF1 knockdown HepG2-NTCP cells were pretreated with 2.5% DMSO for 2 days following HBV infection at an MOI of 200 and were maintained with DMEM containing 2.5% DMSO for 7 days. (A) Knockdown efficiency of was confirmed by WB. (B) Levels of HBV DNA in supernatants were determined by qPCR (% of shCtrl). Secreted HBeAg and HBsAg levels were determined by ELISA (% of shCtrl). (C) Subcellular levels of HBV RNAs in cytoplasm and nucleus were determined by qPCR (% of shCtrl). Graphs show mean ± SD. *p < 0.05; **p < 0.01; ***p < 0.001. (D) The binding of NXF1-HBV RNA, and NXF-HSP70 were detected by RIP assay. (E) Levels of ELAVL1, and CRM1 were determined by WB. (F) ELAVL1 knockdown HepG2-NTCP cells were pretreated with 2.5% DMSO for 2 days following HBV infection at an MOI of 200 and were maintained with DMEM containing 2.5% DMSO for 7 days. Levels of NXF1 were evaluated by WB.
(TIF)

**S7 Fig. ELAVL1 is the adaptor that mediated CRM1 nuclear export pathway for HBV RNAs.** (A-C) The HepG2-NTCP-K7 cells were infected with lentivirus expressing ELAVL1, NXF3, or EIF4E targeted shRNA. The cells were infected with HBV at an MOI = 200 and were harvested at 7 days post HBV infection. (B-D) The HepAD38 cells were infected with lentivirus expressing ELAVL1, NXF3, or EIF4E targeted shRNA. The cells were harvest at 2 days post infection. (A and D) Knockdown efficiency was confirmed by qPCR (% of shCtrl). (B and E) The levels of HBV DNA in culture supernatant were determined by qPCR. The levels of HBeAg and HBsAg in culture supernatant were determined by ELISA (% of shCtrl). (C and F) The subcellular distribution of HBV RNAs was determined by qPCR (% of shCtrl). Graphs show mean ± SD. *p < 0.05; **p < 0.01; ***p < 0.001.
(TIF)

**S8 Fig. DIS3 RNA exosome degraded nuclear retarded HBV RNAs caused by CRM1 knockdown.** (A and B) CRM1 knockdown HepG2-NTCP cells were transfected with the DIS3 targeted shRNAs and were infected with HBV at an MOI = 200. The cells were maintained in DMEM medium containing with 2.5% DMSO for 7 days. (A) Knockdown efficiency of the DIS3 targeted shRNAs was confirmed by qPCR (% of shCtrl). (B) Subcellular levels of HBV RNAs in cytoplasm and nucleus were determined by qPCR (% of shCtrl). (C and D) DIS3 knockdown HepG2-NTCP cells were transfected with the CRM1 targeted shRNAs and were infected with HBV at an MOI = 200. The cells were maintained in DMEM medium containing with 2.5% DMSO for 7 days. (C) Levels of DIS3 and CRM1 mRNA were determined by qPCR (% of shCtrl). (D) Subcellular levels of HBV RNAs in cytoplasm and nucleus were determined by qPCR (% of shCtrl). Graphs were shown as mean ± SD. *, $p < 0.05$; **, $p < 0.01$; ***, $p < 0.001$.
(TIF)

**S1 Table. Critical commercial kits.**
(PDF)

**S2 Table. Antibodies.**
(PDF)

**S3 Table. Oligonucleotides.**
(PDF)

**S4 Table. Chemicals.**
(PDF)

## Acknowledgments

We thank the staffs at the Research Center for Medicine and Structural Biology of Wuhan University for technical assistance.

## Author Contributions

**Conceptualization:** Yuchen Xia.

**Data curation:** Yingcheng Zheng, Mengfei Wang, Jiatong Yin, Yurong Duan, Chuanjian Wu, Zaichao Xu, Yanan Bu, Jingjing Wang, Quan Chen, Guoguo Zhu, Kaitao Zhao, Rong Hua, Yanping Xu.

**Formal analysis:** Yingcheng Zheng, Mengfei Wang, Jiatong Yin, Yurong Duan, Chuanjian Wu, Zaichao Xu, Yanan Bu, Quan Chen.

**Funding acquisition:** Xiaoming Cheng, Yuchen Xia.

**Investigation:** Yingcheng Zheng, Mengfei Wang, Jiatong Yin, Yurong Duan, Chuanjian Wu, Zaichao Xu, Yanan Bu, Lu Zhang.

**Methodology:** Yingcheng Zheng, Mengfei Wang, Jingjing Wang, Quan Chen, Guoguo Zhu, Kaitao Zhao, Rong Hua, Yanping Xu, Xiyu Hu.

**Project administration:** Yuchen Xia.

**Resources:** Yuchen Xia.

**Supervision:** Yuchen Xia.

**Validation:** Yingcheng Zheng, Mengfei Wang.

**Visualization:** Yingcheng Zheng, Mengfei Wang, Jiatong Yin.

**Writing – original draft:** Yingcheng Zheng, Mengfei Wang.

**Writing – review & editing:** Xiaoming Cheng, Yuchen Xia.

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
