## [Decision Letter · Decision Letter 0]

6 Nov 2023

Dear Dr. Xia,

Thank you very much for submitting your manuscript "Hepatitis B virus RNAs co-opt ELAVL1 for stabilization and CRM1-dependent nuclear export" for consideration at PLOS Pathogens. As with all papers reviewed by the journal, your manuscript was reviewed by members of the editorial board and by several independent reviewers. In light of the reviews (below this email), we would like to invite the resubmission of a significantly-revised version that takes into account the reviewers' comments.

We cannot make any decision about publication until we have seen the revised manuscript and your response to the reviewers' comments. Your revised manuscript is also likely to be sent to reviewers for further evaluation.

Sincerely,

Haitao Guo

Guest Editor

PLOS Pathogens

Patrick Hearing

Section Editor

PLOS Pathogens

Kasturi Haldar

Editor-in-Chief

PLOS Pathogens

orcid.org/0000-0001-5065-158X

Michael Malim

Editor-in-Chief

PLOS Pathogens

orcid.org/0000-0002-7699-2064

Reviewer's Responses to Questions

**Part I - Summary**

Reviewer #1: This is an exhaustive study to establish ELAVL1 and its interacting network as regulators of HBV RNA stability and nuclear export. The authors first identified host proteins associated with HBV pgRNA, followed by pathway analysis to focus on ELAVL1 protein for experimental testing. Knocking down ELAVL1 expression from HBV infected HepG2/NTCP cells reduced extracellular HBV DNA, HBeAg and HBsAg, and intracellular core protein and replicative DNA. That was associated with reduced 3.5-kb HBV RNAs and total HBV RNAs in both nuclear and cytoplasmic fractions. Such reductions in HBV markers can be rescued by reintroducing exogenous ELAVL1 (Figs. S3A-C). Similar effect of ELAVL1 silencing was observed in HepAD38 cell line of stable HBV transfection (less dramatic) and in C57B/6 mice transduced with AAV-1.2mer HBV genome. Further study revealed binding of ELAVL1 to 3.5-kb HBV RNAs via AU-rich elements (AREs). Since CRM1 is a known nuclear export receptor for ELAVL1 and its bound RNAs, and considering a previous publication implicating CRM1 in nuclear export of encapsidated pgRNA, the authors examined and confirmed a role of CRM1 in regulating the cytoplasmic (but not nuclear) abundance of HBV RNAs (the use of leptomycin B and a resistant CRM1 mutant made the story more convincing). Moreover they found a role of protein ligands ANP32A and ANP32B in recruiting CRM1 to the HBV RNA-ELAVL1 complex. In the absence of ELAVL1 nuclear HBV RNAs were found to be degraded by DIS3 and RRP6 RNases. While a previous report suggested a role of NXF1 in pgRNA nuclear export, the authors found that silencing NXF1 also reduced ELAVL1 protein level, thus raising the possibility that NXF1 shRNA worked through ELAVL1 instead. Overall this is a solid work centering on ELAVL1 but with ramifications about upstream and downstream events. But since mutating all the four AREs in pgRNA only reduced pgRNA by about 50% (Fig. 4J), ELAVL1 does not explain everything. That is especially true for subgenomic RNAs (no ARE in 0.7-kb RNA and only one ARE in 2.4- and 2.1-kb RNAs). Thus a role of PRE (post-transcriptional regulatory element) in promoting nuclear export of HBV RNAs, especially the subgenomic ones, probably remains valid.

Reviewer #2: ELAVL1 and its effects on the abundance. Further mechanistic studies led to the model that ELAVL1 interacte with ANP32A, ANP32B which tethers to CRM1 for nuclear export and also protects exome mediated RNA decay. Coherent as the model may look like, the authors' interpretation of the effects of ELAVL1 on HBV life cycle might be over-simplified.

Reviewer #3: The authors have done a thorough investigation in the discovery and characterization of ELAVL1 as a host dependency factor that impacts HBV biology. In particular, they used an unbiased proteomics approach that identified ELAVL1 binding to HBV pregenomicRNA (pgRNA) and through a series of experiments they showed that this host factor affects export of HBV RNAs from the nucleus to the cytoplasm by recruiting the nuclear export receptor CRM1 and by protecting HBV RNAs from degradation. Overall, this study is well-designed and the experiments are well-executed supporting the conclusions. Based on the above comments I would recommend this article for publication after the authors address some points mentioned below.

**Part II – Major Issues: Key Experiments Required for Acceptance**

Reviewer #1: In vivo studies are less artificial and important. For Fig. 3, why was there no data on serum HBV DNA and/or intracellular replicative DNA?

Reviewer #2: As the AU-rich element is wide spread in the transcriptome, the effects of ELAVL1 perturbation may be widespread and there may be indirect effects on other host genes that cause the observed manifestation in various levels of HBV life cycle. Indeed, as a recent review pointed out (PMC7555251), ELAVL1 was shown to regulate many cell-cycle related proteins and knock down of ELAVL1 (PMC9316910) led to reduced proliferation in HepAD38 cells. Did the authors looked into this issue?

There are several additional issues that needs clarification.

1. Have the authors attempted to stain pgRNA using RNAscope together with ELAVL1 to validate the RNA-protein interaction?

2. In Figure 3F, the pgRNA positive rate in the AAV-1.2HBV model is 100% which is astonishing. Have the authors done additional controls to validate its specificity?

3. In most of the RNAscope FISH results, the signal is not puntate. Is it because of the limited resolution of the microscope or too much signal amplification was performed? What fluorescent microscope was used? As the RNAscope assay has many differnt versions, please specifiy the catalog number and the technical detail such as the fluorescent channel used.

4. The authors claims that ELAVL1 mediates a post-trnascriptional regulation on HBV RNA. Although this assertion of plausible, necessary work is still needed to rule out the possibility of trnascriptional regulation. Acutally, this is not impossible considering the multi-faceted influence of ELAVL1 knockdown.

5. The effects of CMLD-2 on viral antigen and pgRNA seem to be weak, especially considering the Ki50 of this compound is 350nM.

Reviewer #3: In Figure 1 the authors used a moderate and transient siRNA approach and the study would have benefited from a clean ELAVL1 CRISPR KO followed by a rescue experiment using ELAVL1 overexpression in the background of the KO cells. They already have the overexpression construct as shown in supplementary figure 3.

CMLD-2 as an ELAVL1 inhibitor has been reported to have anti-tumor activity in multiple cancer cell lines. All the studies here use cancer cell lines (HepG2) and the study would be more relevant if inhibition of ELAVL1 (either genetic or pharmacological) could be validated in primary human hepatocytes. In addition, a more thorough cytotoxicity assay and not only b-actin in WB would be useful and easy to generate in cell lines. Especially, when 30uM of a compound is used in Fig 3D and the reduction of HBc is not as robust.

**Part III – Minor Issues: Editorial and Data Presentation Modifications**

Reviewer #1: 1. The initial screening for host factors interacting with HBV RNAs (Fig. S1 and Fig. 1) was based on pgRNA, which and pcRNA has 3 more AREs than 2.4-kb and 2.1-kb RNAs (by the way, most such AREs are not in the “nontranslated region” of pgRNA. ELAVL1 is supposed to bind to AREs in the nontranslated region of RNA). Furthermore, pgRNA is unique in being encapsidated. Thus one concern is that such an experimental approach may be biased towards the identification of host factors interacting with encapsidated HBV RNAs or longer HBV RNA. Certainly subsequent tests seem to support a role of ELAVL1 in stabilizing total HBV RNAs as well, and the effect of ELAVL1 and CRM1 could be reproduced in core-minus mutant (Fig. 6). In some figures, reduction in HBsAg (mostly reflect 2.1-kb RNA) was striking. But according to Fig. 4L, ELAVL1 binding was 3.5kb RNA > 2.4kb RNA > 2.1-kb RNA, and absent in 0.7-kb RNA. Unfortunately, RNA quantification was PCR-based and can only be assigned to “pgRNA” (which should be both pgRNA and pcRNA due to larger size of pcRNA) and “total RNAs”. Thus the specific effect on 2.4-kb RNA, 2.1-kb RNA, and 0.7-kb RNA was unknown. By the way, in Fig. 4M, the 3’ end of the “preS1 mRNA” and “preS2/S mRNA” should be elongated to the same position as other RNAs.

2. While conceptually the story told in this manuscript makes great sense, individual figures did show variability in quantitative values. Thus in Fig. 1D-F (based on HBV infection of HepG2/NTCP cells and transient transfection with ELAVL1 shRNAs), the effect of ELAVL1 silencing was most striking for intracellular HBV DNA, and least for extracellular HBeAg. The effects on extracellular HBV DNA, HBsAg and HBeAg were less striking on HepAD38 cell line (Fig. S2E) or on HepG2/NTCP cells with stable rather than transient ELAVL1 silencing (Fig. S2B). Although CMLD-2, an ELAVL1 inhibitor, dose-dependently reduced such HBV markers in HepG2/NTCP cells (Figs. 2C-E), the effect was modest (up to 40% reduction at the highest concentration).

3. Only 3 earlier papers on PRE were cited (ref. 8, 29, 30). There should be much more work, such as J. Virol. 1994; 68: 3193-3199; and from another group: Nucleic Acids Res. 1998; 26: 4818-4827. According to the current manuscript, it appears that a role of PRE for HBV RNA export is no longer valid. I am unsure about that, especially considering that mutating all the 4 AREs in pgRNA only reduced its level by about 50%.

Reviewer #2: minor issues:

1. repetition in line 515-566.

2. For RNAscope assay, what fluorescent microscope was used? The resolution seems low.

3.line 837, what is the dosage of AAV-1.2HBV?

Reviewer #3: (No Response)

PLOS authors have the option to publish the peer review history of their article (what does this mean?). If published, this will include your full peer review and any attached files.

Reviewer #1: No

Reviewer #2: No

Reviewer #3: No
---

## [Decision Letter · Decision Letter 1]

2 Jan 2024

Dear Dr. Xia,

Thank you very much for submitting your revised manuscript "Hepatitis B virus RNAs co-opt ELAVL1 for stabilization and CRM1-dependent nuclear export" for consideration at PLOS Pathogens. As with all papers reviewed by the journal as Major Revision, your manuscript was re-reviewed by the previous reviewers. While two reviewers are satisfied with the revision, one reviewer has a few additional comments on the rigor and completeness of your study. Based on the reviews, we are likely to accept this manuscript for publication, providing that you modify the manuscript according to the review recommendations. 

Please prepare and submit your revised manuscript within 45 days. If you anticipate any delay, please let us know the expected resubmission date by replying to this email.

Sincerely,

Haitao Guo

Guest Editor

PLOS Pathogens

Patrick Hearing

Section Editor

PLOS Pathogens

Kasturi Haldar

Editor-in-Chief

PLOS Pathogens

orcid.org/0000-0001-5065-158X

Michael Malim

Editor-in-Chief

PLOS Pathogens

orcid.org/0000-0002-7699-2064

Reviewer Comments (if any, and for reference):

Reviewer's Responses to Questions

**Part I - Summary**

Reviewer #1: The authors have responded to critiques raised in the previous round of review and added some additional data. I have no further comments.

Reviewer #2: In this revised manuscript, the authors addressed some of my comments by providing additional information and experimental evidence to consolidate the conclusion of the article. Nevertheless, I still have concern on the final proposed model on the mechanism of ELAVL1 on HBV RNA. Considering the sheer number of publications pointing to the pleotropic effects of this gene on cell biology (cell cycle, senescence, inflammation), rigorous examination of the other possible explanations should be taken into accoutn to minimize the self-fulfilled tendency of the manuscript.

Reviewer #3: The authors addressed most of the points that the reviewers raised. Incorporating all the methods in the manuscript and the figures in a clear way will be needed during the final stage but I don't have any other comments and based on the revision I recommend that the manuscript to be accepted.

**Part II – Major Issues: Key Experiments Required for Acceptance**

Reviewer #1: none.

Reviewer #2: For example, although the authors used DMSO to stall the division of the cells, this does not translate to the complete absence of ELAVL1-mediated perturbation on the general cellular physiology. The author should at least mitigate this by providing RNA-seq data on the ELAVL1 knocked down cells in the absence or presence of HBV replication. The level of the perturbation of the general transcriptome on ELAVL1 silencing would generate an objective measure of the pleotropic effects of this gene.

Also, there are a number of vulnerabilities in the presented data that are difficult to reconcile with the theory. For instance, the limited effects of CMLD-2 on the HBV transcripts and protein expression are an obvious counterargument for the proposed theory. Unfortunately, the authors chose to tune down this finding.

Second, the S/E antigen level in the animal experiment (Fig. 3) showed a very late effect (3-4 weeks) on the ELAV1 knock down on the viral antigen. This seems to be at odds with the cell culture results, in which the effects on the viral RNA and antigen can be shown in days. Is there good explanation for this?

Reviewer #3: no major issues

**Part III – Minor Issues: Editorial and Data Presentation Modifications**

Reviewer #1: none.

Reviewer #2: A minor issue.

The authors heavily used qRT-PCR to quantify the effects of ELAVL1 knockdown on different viral transcript.

Some additional Northern blot in Figure 2 would be much more reassuring.

Reviewer #3: no minor issues but the figures in the revision had very low resolution.

PLOS authors have the option to publish the peer review history of their article (what does this mean?). If published, this will include your full peer review and any attached files.

Reviewer #1: No

Reviewer #2: No

Reviewer #3: No

Figure Files:

Data Requirements:

Reproducibility:

References:

---

## [Editor Report · Decision Letter 2]

25 Jan 2024

Dear Dr. Xia,

We are pleased to inform you that your manuscript 'Hepatitis B virus RNAs co-opt ELAVL1 for stabilization and CRM1-dependent nuclear export' has been provisionally accepted for publication in PLOS Pathogens.

Best regards,

Haitao Guo

Guest Editor

PLOS Pathogens

Patrick Hearing

Section Editor

PLOS Pathogens

Michael Malim

Editor-in-Chief

PLOS Pathogens

orcid.org/0000-0002-7699-2064
---

## [Editor Report · Acceptance letter]

30 Jan 2024

Dear Dr. Xia,

We are delighted to inform you that your manuscript, " Hepatitis B virus RNAs co-opt ELAVL1 for stabilization and CRM1-dependent nuclear export ," has been formally accepted for publication in PLOS Pathogens.

Best regards,

Michael Malim

Editor-in-Chief

PLOS Pathogens

orcid.org/0000-0002-7699-2064